# The clinical-stage drug BTZ-043 accumulates in murine tuberculosis lesions and efficiently acts against *Mycobacterium tuberculosis*

Andreas Römpp [1,2] ✉, Axel Treu[1,2,14], Julia Kokesch-Himmelreich[1,2,14], Franziska Marwitz[3,4,14], Julia Dreisbach [2,5], Nadine Aboutara[3,4], Doris Hillemann[6], Moritz Garrelts[4,7], Paul J. Converse [8], Sandeep Tyagi[8], Sina Gerbach[9], Luzia Gyr [10], Ann-Kathrin Lemm[4,7], Johanna Volz[7], Alexandra Hölscher[7], Leon Gröschel [1,2], Eva-Maria Stemp[1,2], Norbert Heinrich[2,5,11], Florian Kloss[9], Eric L. Nuermberger [8], Dominik Schwudke [3,4,12], Michael Hoelscher [2,5,11,13], Christoph Hölscher [4,7] & Kerstin Walter [4,7] ✉

The development of granulomas with central necrosis harboring *Mycobacterium tuberculosis* (Mtb) is the hallmark of human tuberculosis (TB). New anti-TB therapies need to effectively penetrate the cellular and necrotic compartments of these lesions and reach sufficient concentrations to eliminate Mtb. BTZ-043 is a novel antibiotic showing good bactericidal activity in humans in a phase IIa trial. Here, we report on lesional BTZ-043 concentrations severalfold above the minimal-inhibitory-concentration and the substantial local efficacy of BTZ-043 in interleukin-13-overexpressing mice, which mimic human TB pathology of granuloma necrosis. High-resolution MALDI imaging further reveals that BTZ-043 diffuses and accumulates in the cellular compartment, and fully penetrates the necrotic center. This is the first study that visualizes an efficient penetration and accumulation of a clinical-stage TB drug in human-like centrally necrotizing granulomas and that also determines its lesional activity. Our results most likely predict a substantial bactericidal effect of BTZ-043 at these hard-to-reach sites in TB patients.

Tuberculosis (TB) is caused by *Mycobacterium tuberculosis* (Mtb), affects about 10 million people globally each year and was responsible for 1.3 million deaths in 2022. Treatment of drug sensitive TB currently requires either a six-month regimen of pyrazinamide (PZA), isoniazid (INH), rifampicin (RIF) and ethambutol or the recently recommended four-month regimen of INH, PZA, rifapentine and moxifloxacin (MXF)[1]. While some of these first-line drugs have been in use for over 50 years, the emergence of drug resistant Mtb strains urgently requires new anti-TB drugs[2]. Hence, efficient pre-clinical screening and testing of antimycobacterial candidates is essential for the rapid development of novel therapeutics[3].

The histopathological hallmark of human TB are centrally necrotizing granulomas which are composed of a lipid-rich, non-vascularized necrotic core adjacent to a rim of foamy macrophages and are surrounded by a fibrous capsule and a layer of epitheloid macrophages. In these necrotic lesions, extracellular Mtb can reside in a metabolically dormant stage, well protected from the immune system[4]. Furthermore, the diffusion of antibiotics into these lesions may be hampered due to their complex structure. However, the ability of drugs to penetrate necrotic granulomas significantly affects the efficacy of treatment[5]. To predict this penetration capability into centrally necrotizing granulomas it is essential to exploit animal

models that reflect the specific human TB pathology as much as possible[6,7]. Regular inbred mice (BALB/c or C57BL/6) are the most commonly used models for pre-clinical testing, but pulmonary granulomas of these mice lack the non-vascularized caseous necrotic center observed in humans[8], which is the compartment that is most difficult to treat. Hence, advanced mouse models such as C3HeB/FeJ[9] or interleukin (IL)-13-overexpressing (tg)[10] mice, which reflect the pathology of human TB in many aspects, are increasingly used for an improved pre-clinical evaluation of new drugs by assessing their potential to penetrate centrally necrotizing granulomas and thus predict their local antimycobacterial activity[11–13].

The distribution of endogenous and exogenous compounds in tissue sections can be analyzed by MALDI imaging[14]. By combining the specificity of mass spectrometry with spatial information this technique is ideal to investigate the localization of compounds in tissue samples and has been used for the analysis of anti-TB drugs, in particular by the Dartois group[5,15,16]. A reliable identification of drug compounds in tissue requires high mass resolution and mass accuracy[14]. The necessary spatial resolution depends on the species and the targeted histological structures. For the murine model used in this study a spatial resolution in the range of 10 μm is recommended in order to resolve sublesional structures of necrotic granulomas. We have recently established a method that combines these two features for the analysis of anti-TB drugs in lung tissue of Mtb-infected mice[12].

A drug target on which the greatest hopes are currently pinned is the decaprenylphosphoryl-β-D-ribose oxidase (DprE1) enzyme, which is essential for the biosynthesis of the mycobacterial cell wall[17]. There are currently 4 DprE1 inhibitors in clinical development. BTZ-043 and PBTZ169 bind covalently to DprE1 whereas TBA-7371 and OPC167832 bind in a noncovalent manner[18–21]. BTZ-043 and OPC167832 have successfully completed early bactericidal activity phase IIa studies over 14 days where safety and efficacy have been evaluated in TB patients[22]. Both compounds, OPC-167832 by Otsuka and BTZ-043 by University Hospital, University of Munich (LMU), have been selected by large international clinical trials networks (PanTB[23] and UNITE4TB[24]) and will be evaluated clinically in advanced phase IIb combination studies.

BTZ-043 was first described in 2009 as the lead compound of the novel class of benzothiazinones, which are active against Mtb[18]. Benzothiazinones are nitroaromatic compounds which inhibit the cell wall formation by covalent binding to Cys387 in the catalytic pocket of DprE1, thus blocking the formation of decaprenylphosphoryl-β-D-arabinose which is the sole precursor for the synthesis of the arabinan moiety of cell wall arabinogalactan and lipoarabinomannan[25–28]. The lack of these polysaccharides disturbs the integrity of the mycobacterial cell wall and induces cell lysis. BTZ-043 shows a high antimycobacterial activity with minimal inhibitory concentrations (MIC) against H37Rv and clinical multidrug or extremely drug resistant isolates in the nanomolar range[18,29]. It is highly active against intracellular mycobacteria in macrophages and has a good efficacy in the standard mouse model of TB[18,19]. However, the lowest dose achieving maximum efficacy, the sublesional distribution and local lesional activity of BTZ-043 in an advanced pre-clinical TB mouse model that recapitulates key elements of human TB pathology have not been studied so far. These aspects are important for the design of clinical studies and are instrumental to predict the antimycobacterial effect under physiological conditions in humans.

In this study, we therefore comprehensively evaluate BTZ-043 in standard and advanced pre-clinical TB mouse models by combining microbiological and molecular efficacy studies. This includes the determination of pulmonary and lesional drug concentration by laser-capture microdissection (LCM), subsequent LC-MS/MS and quantitative Mtb 16S rRNA expression, and the assessment of drug penetration into centrally necrotizing granulomas by high spatial resolution MALDI imaging. Our study visualizes an early accumulation and determines the local efficacy of a clinical-stage TB drug in human-like granulomatous lesions resulting in potent antimycobacterial activity.

## Results

### Dose and time dependent antimycobacterial activity of BTZ-043

To determine the maximum effective dose and the pharmacokinetic (PK) parameter correlated with the efficacy of BTZ-043, we tested several dose levels and dosing frequencies in Mtb-infected BALB/c mice and assessed colony forming units (CFU) reduction and plasma PK levels prior to analyzes in the IL-13[tg] advanced mouse model.

In a dose escalation study, treatment of BALB/c mice with 5 different doses of BTZ-043 (50 mg/kg/day to 1000 mg/kg/day microcrystalline BTZ-043) started 3 weeks after infection and CFU reduction was determined after 4, 6 or 8 weeks of therapy. Mice receiving the vehicle suspension served as a negative control, whereas INH treatment (25 mg/kg/day) served as a positive control. During the early stage of infection an exponential growth of mycobacteria was observed which was controlled after 3 weeks resulting in a plateau of bacterial burden in mice treated with the vehicle suspension whereas a reduction of the bacterial load was observed in all BTZ-043 treatment groups (Fig. 1a). After 4 weeks of therapy, BTZ-043 was active at all doses investigated and the bacterial burden was significantly reduced by > 0.7 $\log_{10}$ CFU in comparison to the vehicle control group (Fig. 1b). When administered for 6 weeks, BTZ-043 at 250 mg/kg/day, 500 mg/kg/day and 1000 mg/kg/day had significantly greater antimycobacterial activity than the first-line drug INH while lower doses displayed a comparable (100 mg/kg/day) or reduced (50 mg/kg/day) activity. The significantly increased efficacy of the 3 highest BTZ-043 doses compared to INH therapy was also observed after 8 weeks of treatment. Furthermore, the pulmonary bacterial burden of these treatment groups was more than 2.4 $\log_{10}$ CFU lower than the vehicle control group. Consequently, these results demonstrate that BTZ-043 has a dose and time dependent bactericidal activity. As the efficacy did not significantly increase beyond 250 mg/kg/day, this was defined as the lowest dose with maximum effect (LME) for BTZ-043 in the mouse model of TB.

To assess the evolution of resistance to BTZ-043, mycobacteria from BTZ-043 treated mice, untreated mice and H37Rv (control), were plated on agar containing increasing concentrations of BTZ-043 (1, 5, and 10 ng/mL). Under the conditions tested, 1 ng/mL BTZ-043 did not restrict mycobacterial viability since the growth of bacteria isolated from BTZ-043 treated mice, untreated mice and H37Rv was equally robust. No growth was observed in the presence of 5 or 10 ng/mL BTZ-043. Thus, the selection of drug-resistant mutants during treatment can be excluded. Furthermore, mutations in the *dprE1* (*rv3790*) gene and the *mmpR5* (*rv0678*) gene have been reported to confer BTZ-043 resistance[18,30,31]. Therefore, the DNA sequences of both genes were determined from Mtb that were isolated from representative BTZ-043 treated mice. No alterations from the wildtype Mtb sequence were detected which is in accordance with the microbiological drug susceptibility results.

In the next experiment, BALB/c mice were used to determine the PK parameter driving the antimycobacterial activity of BTZ-043. We compared two different formulations, one suspension containing microcrystalline BTZ-043 which is hardly soluble in carboxymethyl cellulose (CMC) / Tween 80 and an experimental amorphous preparation with improved dissolution behavior. For comparison of the PK/PD characteristics in vivo, both formulations were administered in parallel. Total daily doses of 2.5, 5, 10, 20, 25, 50 and 100 mg/kg of BTZ-043 amorphous preparations were administered either once (QD) or twice (BID) daily as well as 250 mg/kg (QD) of BTZ-043 microcrystalline formulation for a total of 6 weeks. For amorphous BTZ-043, the detection of the pulmonary bacterial burden revealed a dose dependent decrease of CFUs (Fig. 1c). For doses representing the steep portion of the dose-response curve (5 to 50 mg/kg/day), fractionation

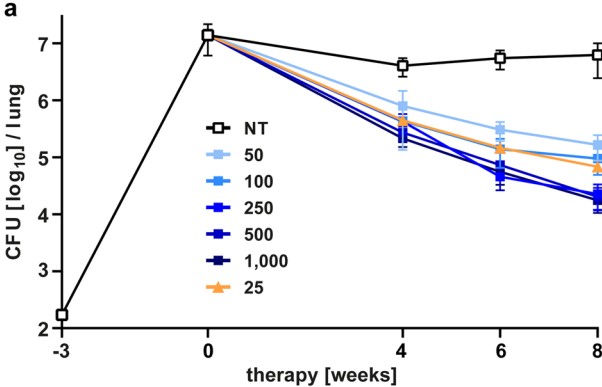

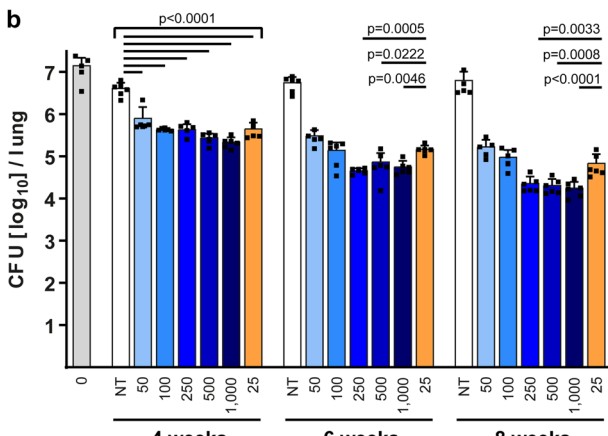

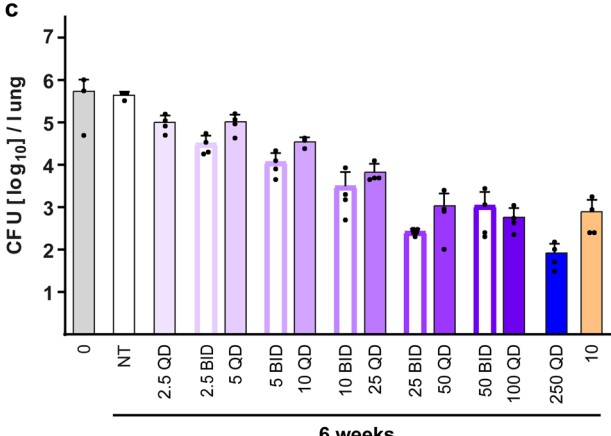

**Fig. 1 | Efficacy of BTZ-043 in Mtb-infected BALB/c mice. a** and **b** BTZ-043 dose escalation study in the chronic model of TB. Female BALB/c mice were infected by aerosol with an average of 171 CFU Mtb H37Rv 3 weeks before treatment (−3). The pulmonary bacterial burden was assessed at the start of therapy (0) and after 4, 6 and 8 weeks of treatment with the indicated doses [mg/kg/day] of microcrystalline BTZ-043 (blue colors), while isoniazid (INH, 25 mg/kg/day, orange color) and vehicle (NT, white color) served as controls. Results are expressed as mean and SD ($n = 5$ mice per group for time points (-3), (0) and 8 weeks of treatment with 50 mg/kg BTZ-043, $n = 6$ mice for all other drug treatment groups; two-way ANOVA with Bonferroni's posttest). **c** BTZ-043 dose-fractionation study. Female BALB/c mice were infected with an average of 20 CFU Mtb H37Rv 3 weeks before treatment started with different doses [mg/kg] of amorphous BTZ-043 (purple bars) administered either once (QD, filled bars) or twice (BID, open bars) daily or with microcrystalline BTZ-043 (QD, blue bar). INH therapy (10 mg/kg/day, light orange bar) or untreated mice (NT, white bar) served as controls. Treatment efficacy was determined after 6 weeks of therapy. Data represent mean and SD ($n = 3$ per group for the time point (0), the NT group and the 10 mg/kg BTZ-043 QD treatment group, $n = 4$ mice for all other treatment groups; one-way ANOVA with Bonferroni's posttest for comparison of same total doses administered either BID or QD). Source data are provided as Source Data file.

formulation, the PK profile of the two formulations was compared at this dose, revealing nearly identical $C_{max}$ and AUC, but displaying slightly different kinetics (Fig. 2a, b).

In combination, these findings suggest a clear dose-dependent bactericidal activity until BTZ-043$_{total}$ exposures reach saturation at around 8000 ng/mL for $C_{max}$ and 30,000 h·ng/mL for AUC, respectively. Furthermore, the fractionation effect disappeared at doses above 50 mg/kg/day while efficacy still increased with higher doses up to 250 mg/kg/day. Therefore, we have chosen the 250 mg/kg/day dose (QD) as the optimal treatment dose for the evaluation of the efficacy and penetration capability of BTZ-043 into centrally necrotizing granulomas of Mtb-infected IL-13$^{tg}$ mice.

## Bactericidal activity of BTZ-043 under conditions of centrally necrotizing granulomas

The goal of TB therapy is to reduce the bacterial burden in lung tissue. While some anti-TB drugs are highly effective in standard mouse models that develop non-necrotic granulomas upon Mtb infection, their activity is reduced in advanced mouse models with necrotic granulomas[32], probably due to their limited diffusion into necrotic lesions[12,13]. IL-13$^{tg}$ mice develop a human-like pathology after Mtb infection[10]. Consequently, they are ideal to investigate the anti-mycobacterial activity of BTZ-043 under conditions of centrally necrotizing granulomas in combination with drug penetration into these lesions and pulmonary drug concentration.

Therapy of Mtb-infected IL-13$^{tg}$ mice started after the development of necrotic granulomas had been histologically confirmed (Supplementary Fig. 1). BTZ-043 was administered at the previously determined LME (250 mg/kg/day) for a total of 10 consecutive days as the major aim was the detection of drug distribution within centrally necrotizing granulomas and pulmonary drug concentration under steady state conditions. At the start of treatment pulmonary mycobacterial counts were 8.55 log10 CFU and only 10 days of BTZ-043 therapy reduced the bacterial load significantly by 0.9 log10 CFU (Fig. 3a). The generation of resistance to BTZ-043 was not observed as Mtb from drug treated IL-13$^{tg}$ mice did not grow on agar plates containing 5 or 10 ng/mL BTZ-043. Since in IL-13$^{tg}$ mice, most pulmonary mycobacteria reside in highly stratified centrally necrotizing granulomas[10,13] and BTZ-043 treatment reduced the bacterial burden by 86% (from 35.1 × 10$^7$ before treatment to 4.8 × 10$^7$ after treatment), it is reasonable to assume that BTZ-043 reaches its mycobacterial target within TB lesions. Therefore Ziehl-Neelsen (ZN) staining was performed to detect acid fast bacilli (AFB) within centrally necrotizing granulomas. As expected, high amounts of mycobacteria were

of the dose into two administrations showed a modest, but consistent, trend towards increased mycobactericidal activity compared to once daily administration at identical total doses. The highest bactericidal activity was again observed at 250 mg/kg daily dose (microcrystalline BTZ-043) and the CFU reduction compared to the untreated control was 3.7 log10 CFU.

Next, we assessed the PK properties of BTZ-043 for different doses and formulations in plasma samples of naïve BALB/c mice (see Supplementary Table 1 for treatment and blood sampling scheme). BTZ-043 was rapidly absorbed, with a $T_{max}$ of 0.5–1 h (Fig. 2a, b; Supplementary Data 1). In the dose range of 2.5–50 mg/kg/day, dose-proportional increases of BTZ-043$_{total}$ plasma $C_{max}$ and AUC were observed with 520 to 8070 ng/mL and 666 to 14,700 h·ng/mL, respectively. At 250 mg/kg/day, $C_{max}$ was not further increased, and AUC increased less than dose proportionally (Fig. 2b). Since 250 mg/kg/day was identified as the LME for the microcrystalline

**a**

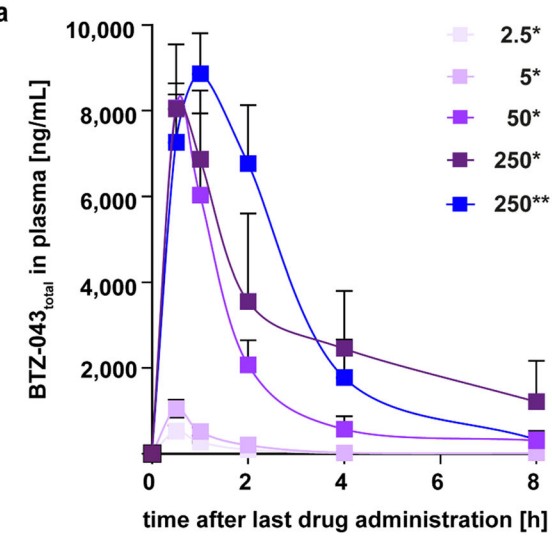

**b**

| Dose [mg/kg] | $C_{max}$ [ng/mL] | $T_{max}$ [h] | $AUC_{inf}$ [h·ng/mL] |
|---|---|---|---|
| 2.5* | 520 | 0.5 | 666 |
| 5* | 1,050 | 0.5 | 1,360 |
| 50* | 8,070 | 0.5 | 14,700 |
| 250* | 8,040 | 0.5 | 31,000 |
| 250** | 8,860 | 1.0 | 27,000 |

**Fig. 2 | Pharmacokinetic profile of BTZ-043$_{total}$ in mice.** Six naïve BALB/c mice per group were treated with amorphous formulation (*) or microcrystalline formulation (**) of BTZ-043 at indicated doses [mg/kg] for 5 days by oral gavage. **a** Plasma concentration-time curves (mean and SD, $n = 3$ mice per timepoint). **b** PK parameters (mean, $n = 3$ mice per timepoint). Source data are provided as Source Data file and as Supplementary Data 1.

observed in the rim of foamy macrophages and in the necrotic core of untreated mice (Fig. 3b, left panel). In comparison, BTZ-043 treated mice showed the most notable reduction in AFB within the layer of foamy macrophages (Fig. 3b, middle and right panel). Consequently, not only an overall reduction of the pulmonary bacterial burden was observed in BTZ-043 treated IL-13$^{tg}$ mice but also remarkably less AFB within the macrophage zone of centrally necrotizing granulomas. A change in granuloma size or composition due to BTZ-043 treatment was not observed.

A high efficacy of BTZ-043 was also observed during prolonged therapy as drug treatment of Mtb-infected IL-13$^{tg}$ mice for a longer period of time reduced the pulmonary bacterial burden by more than 2 log$_{10}$ CFU compared to the vehicle group (Fig. 4a). In order to assess whether BTZ-043 exerts also its antimycobacterial activity in the necrotic center of granulomas, a defined area of this region was excised using LCM (Fig. 4b, left panel) and the bacterial load per area was determined by a molecular-bacterial-load assay (MBLA) based on the expression of Mtb 16S rRNA as described[33]. By 2 weeks of treatment, this expression was already decreased in the BTZ-043 treated animals compared to the vehicle group (Fig. 4b, right panel). In the further course of therapy, the bacterial load in the necrotic center was strongly reduced in drug treated mice. Taken together BTZ-043 reduces the overall pulmonary CFU counts in IL-13$^{tg}$ mice and most importantly exerts its antimycobacterial activity in centrally necrotizing granulomas.

**In vitro permeability coefficients of BTZ-043**

Centrally necrotizing granulomas are compact structures with a rim of foamy macrophages surrounding a necrotic core which is devoid of vascularization so that the supply of antibiotics from the blood stream is impaired. Consequently, penetration and distribution of drugs in these lesions mainly depends on diffusion kinetics. BTZ-043 is a nonpolar lipophilic substance that is not significantly charged under physiological conditions. We therefore assumed that transport of BTZ-043 through epithelial barriers and within tissue is mainly driven by passive diffusion. Cell permeation is regularly determined in the Caco-2 assay, which serves as a highly standardized method to predict intestinal uptake of drugs[34]. Thus, we reasoned that Caco-2 permeability could serve as a simplified readout to rank permeabilities of anti-TB drugs also regarding their potential to penetrate necrotic granulomas. The Caco-2 permeability was previously reported for a

range of anti-TB drugs[35], but not for BTZ-043. Therefore, the rate of flux across polarized Caco-2 cell monolayers was determined for BTZ-043. The permeability coefficient ($P_{app}$) of BTZ-043 was $9.44 – 11.28 \times 10^{-6}$ cm/s from apical to basolateral (AB) side or $6.90 – 12.91 \times 10^{-6}$ cm/s from basolateral to apical (BA) side (Supplementary Fig. 2, Supplementary Table 2; Supplementary Data 5). BTZ-043 can therefore be classified as a permeable drug[35,36]. Efflux ratios for BTZ-043 also indicate that passive diffusion is dominant over active transport mechanisms (Supplementary Table 2).

It has been previously shown that the highly permeable drug PZA ($P_{app}$: $70.4 \times 10^{-6}$ cm/s (AB) and $37.1 \times 10^{-6}$ cm/s (BA))[35] readily penetrates centrally necrotizing granulomas[5,11-13] whereas the low permeability drug clofazimine ($P_{app}$: $0.17 \times 10^{-6}$ cm/s (AB) and $0.18 \times 10^{-6}$ cm/s (BA))[35] shows only very limited penetration[5,12,13].

The permeability of BTZ-043 is in the range of fluoroquinolones and linezolid ($8 – 18 \times 10^{-6}$ cm/s)[35], which show some variation regarding their ability to penetrate centrally necrotizing granulomas[5,16,37]. Therefore, we performed MALDI imaging in order to specify the distribution pattern of BTZ-043 within the complex structure of centrally necrotizing granulomas in vivo, as described in the following section.

**Early BTZ-043 accumulation in the rim of foamy macrophages**

Since the distribution of antibiotics in Mtb-infected lungs is often drug-specific and depends on the type of lesion[5,13], MALDI imaging was used to analyze the ability of BTZ-043 to penetrate centrally necrotizing granulomas. To this end we utilized our recently established infrastructure for an improved and accelerated evaluation of novel anti-TB drug candidates[12,13].

MALDI imaging outside a biosafety level 3 (BSL-3) facility requires the inactivation of mycobacteria, which can be achieved by γ-irradiation[13] or heat deactivation[38,39]. We have recently established a workflow based on γ-irradiation which is compatible with high-resolution MALDI imaging[12]. This protocol allows uninterrupted cooling of the tissue section, which is necessary to prevent leaking of certain anti-TB drugs[40]. Since γ-irradiation can potentially cause degradation of compounds[13] we carefully evaluated our protocol for BTZ-043 stability by using a mimetic tissue model[41] as well as lung tissue from BTZ-043 treated mice. We could show that the dosage of γ-irradiation used in our study does not impact the concentration and distribution of BTZ-043 neither within liver mimetic cryosections nor within lung tissue cryosections (Supplementary Fig. 3; Supplementary Data 6, 7 and 8).

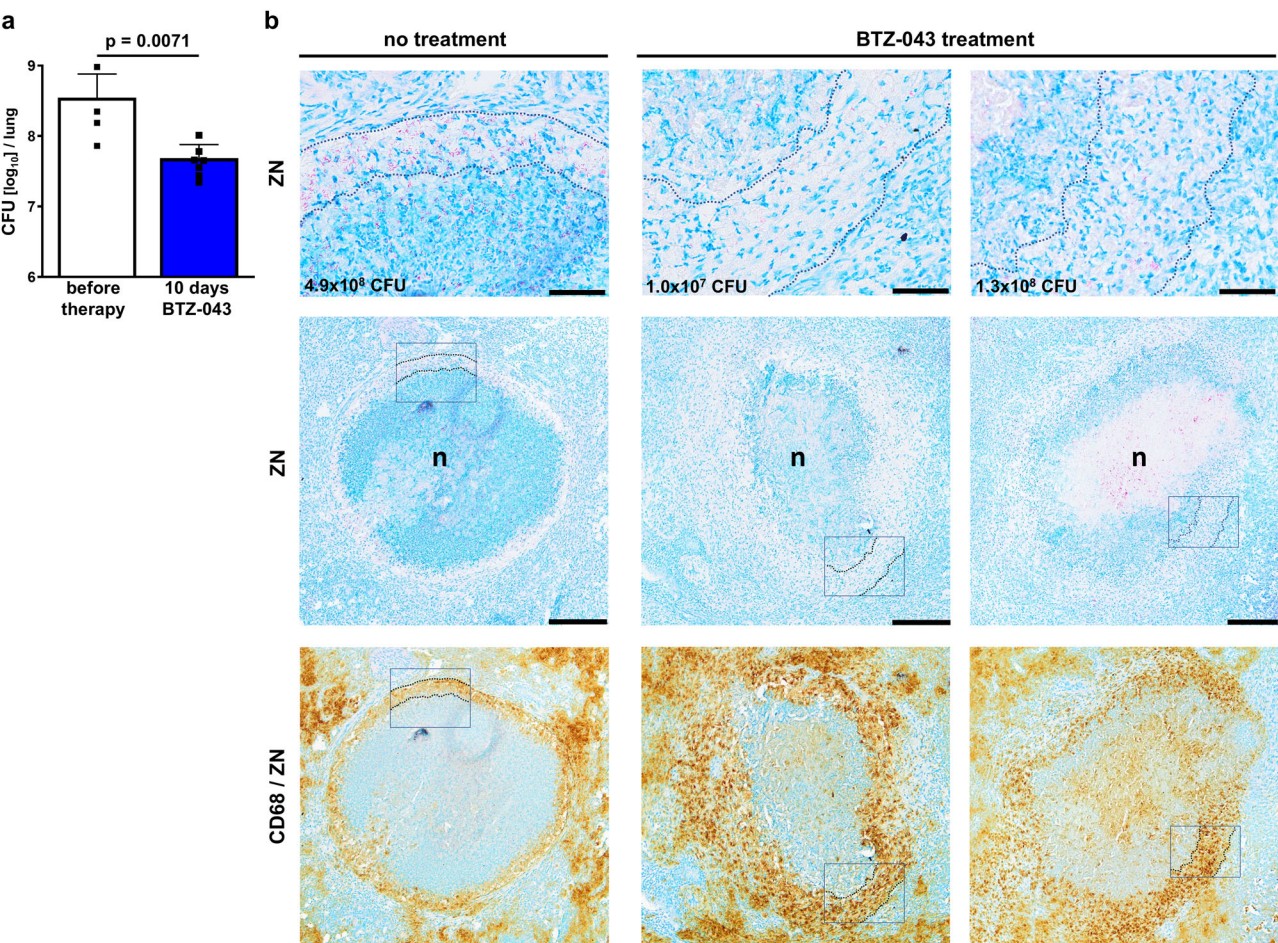

**Fig. 3 | Early antimycobacterial activity of BTZ-043 under conditions of centrally necrotizing granulomas. a** Bacterial burden in lungs of IL-13[tg] mice. Animals were infected with an average of 263 CFU Mtb H37Rv via the aerosol route. Nine weeks after infection, mice were treated with a daily dose of 250 mg/kg of BTZ-043 for 10 days. The pulmonary bacterial burden was determined before and after therapy (8.55 $\log_{10}$ CFU and 7.69 $\log_{10}$ CFU, respectively). Data represent mean and SD ($n = 4$ mice per group before therapy, $n = 7$ mice per group for BTZ-043 treatment) and statistical analysis was performed by an unpaired student's $t$ test (two-tailed) on $\log_{10}$-transformed CFU data. One experiment representative of 2 is shown. Source data are provided as Source Data file. **b** Detection of acid fast bacilli in centrally necrotizing granulomas of untreated or BTZ-043 treated mice by Ziehl-Neelsen (ZN) staining. Animals were infected by aerosol with an average of 106 CFU Mtb H37Rv and treatment with 250 mg/kg/day of BTZ-043 for 10 days started 9 weeks after infection. Upper panels: higher magnifications of ZN-stained specimens with rimmed zones of foamy macrophages. Total pulmonary CFU counts determined for the respective mouse are shown in the lower left corner. Middle panels: lower magnifications of the same ZN-stained specimens depicting the entire granulomas and areas of higher magnification displayed in the upper panels are outlined. Lower panels: subsequent CD68/ZN staining of the same sections for the detection of foamy macrophages. Images are representative of 2 independent experiments. Scale bars: upper panel 50 μm, middle and lower panel 200 μm.

For the correlation of lesion pathology with drug distribution, lung cryosections, obtained from an Mtb-infected IL-13[tg] mouse that was sacrificed 2 h post-dose, were histologically characterized. HE staining revealed two highly stratified granulomas with an eosinophilic necrotic center and an overall size of 0.7–0.9 mm in diameter (Fig. 5a). Both granulomas were surrounded by a prominent fibrous capsule (Fig. 5b). A rim of foamy macrophages, which stains less intensely with eosin but positive for CD68 (Fig. 5c) and is enriched in lipid droplets (Fig. 5d), was detected next to the collagen encapsulation in both granulomas, thus strongly resembling foamy macrophages observed in human granulomas[42].

MALDI imaging of BTZ-043 was performed with a pixel size of 10 μm on a neighboring cryosection. BTZ-043 was detected throughout the tissue section, with a higher abundance in the granuloma areas (Fig. 5e; Supplementary Data 2 and 8), see Supplementary Figs. 4–6 for a single pixel spectrum, on-tissue MS/MS, and a negative control. The correlation with the histological staining of adjacent sections given in Fig. 5a–d revealed that the area of high BTZ-043 abundance is located inside the fibrous capsules of the

granulomas and coincides with the cellular compartment mainly consisting of foamy macrophages. Using our in-house penetration analysis tool[12], BTZ-043 penetration plots were generated for both granulomas (Fig. 5f). A data point in these plots represents the average intensity of BTZ-043 in all pixels with the same distance to the granuloma edge (0 μm on $x$ axis). The granuloma edges (depicted in Fig. 5e as blue and purple lines) were determined based on co-detected ions (for more details see Supplementary Fig. 7). Here, the determined granuloma edges are located between the fibrous capsule and the zone of foamy macrophages. The right granuloma is sectioned in its central region showing a prominent necrotic core. The left granuloma is sectioned in a more peripheral region so that the cellular layer, which mainly consists of foamy macrophages, is more prominent. A neutrophil-rich zone between the macrophage layer and the necrotic core is present in both granulomas. This complex structure is reflected in the penetration plots of BTZ-043. In both granulomas, the BTZ-043 intensity shows a steep increase directly inside the edge i.e., the macrophage layer. Due to the thicker macrophage layer in the left granuloma, however, BTZ-043

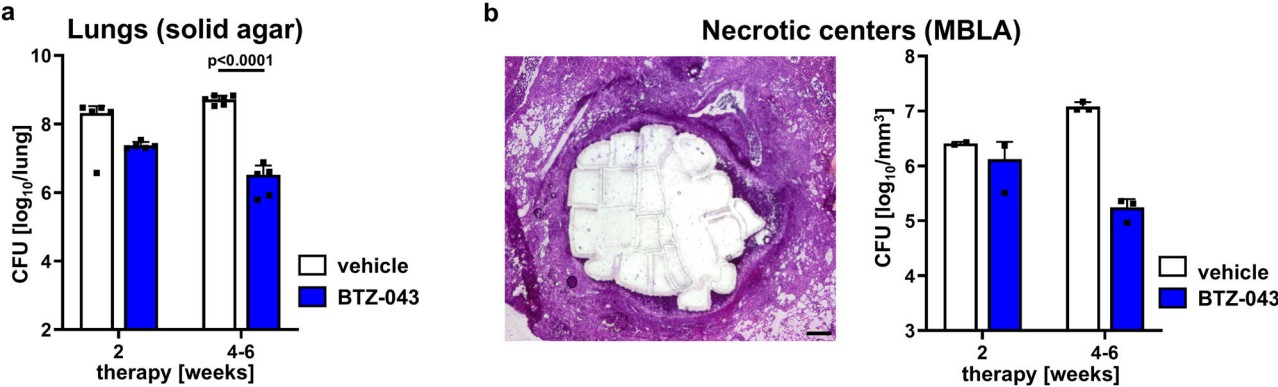

**Fig. 4 | Pulmonary and lesional efficacy of BTZ-043 over the course of treatment.** Animals were infected with an average of 341 CFU Mtb H37Rv via the aerosol route. Ten weeks after infection, mice were treated 5 times a week with 250 mg/kg of BTZ-043 or the vehicle control. After 2 weeks and between 4 and 6 weeks, lungs were removed to determine the bacterial load in the lung and in the necrotic center of granulomas. Since some mice receiving the vehicle suspension became moribund, the second analysis time point comprises a treatment period of 4 to 6 weeks. **a** Pulmonary bacterial burden determined by CFU counts on solid agar medium. Data represent mean and SD ($n = 5$ mice per group for BTZ-043 treatment groups and the group receiving vehicle for 2 weeks, $n = 6$ mice for the group receiving vehicle for 4 to 6 weeks), and statistical analysis was performed by an unpaired student's $t$ test (two tailed) on $\log_{10}$-transformed CFU data. **b** Lesional efficacy of BTZ-043. Left panel: Visual representation of necrotic area collected by LCM. Scale bar: 200 μm. Right panel: The local bacterial load was quantified by 16S rRNA gene expression of Mtb using a molecular bacterial load assay (MBLA) after LCM of the necrotic core of granulomas on serial lung sections normalized to the volume of the isolated area. Data represent mean and SD ($n = 2$ mice per group for 2 weeks of therapy, $n = 3$ mice per group for 4 to 6 weeks of therapy; two experiments were conducted: 1 experiment with an analysis time point after 2 weeks of treatment and 1 repeat experiment in which samples were obtained 2 weeks and between 4 and 6 weeks of treatment; the results of the local bacterial loads in the two experiments were similar). Source data are provided as Source Data file.

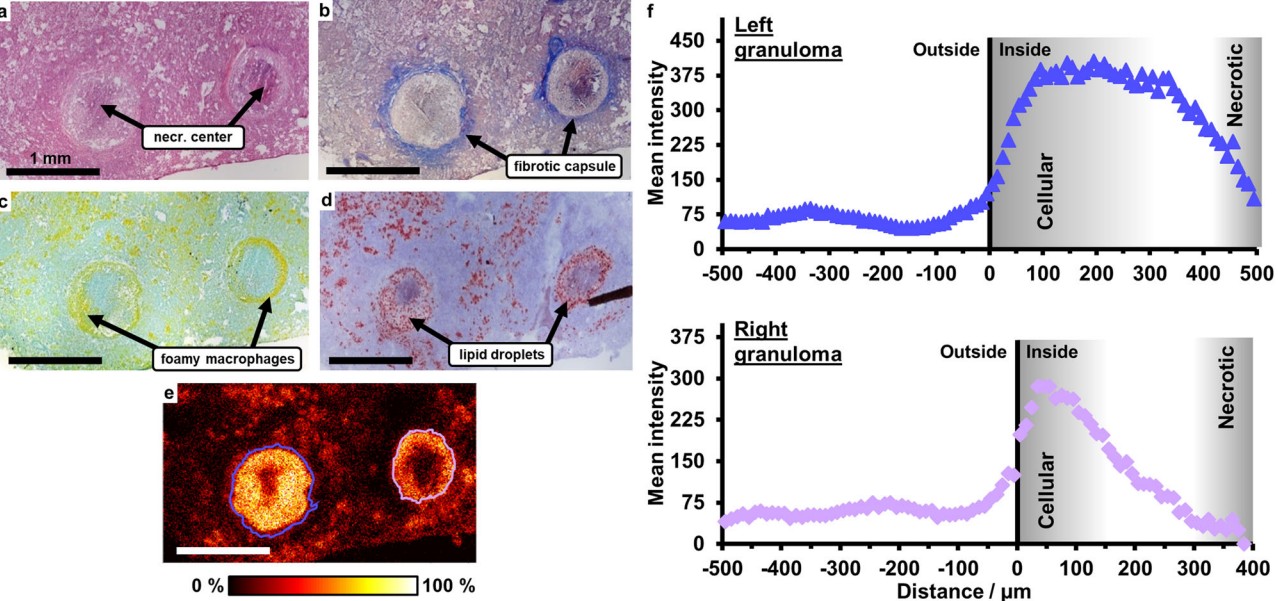

**Fig. 5 | Correlation of lesion pathology and BTZ-043 penetration.** Treatment with 250 mg/kg/day of BTZ-043 for 10 days started 9 weeks after infection with 263 CFU Mtb H37Rv. Lung tissue of an IL-13[tg] mouse was collected 2 h after the last administration and serial cryosections were prepared for histological characterization of lesions and MALDI imaging. **a** HE-stained specimen. **b** Trichrome-stained specimen. **c** CD68/ZN-stained specimen. **d** Oil Red O-stained specimen. **e** Ion image of BTZ-043 [M + H]⁺ $C_{17}H_{17}F_3N_3O_5S^+$, $m/z$ 432.08355, 10 μm pixel size. The blue and purple lines indicate the determined granuloma edges. The stainings given in **a,** **b** and **c** were performed on sections directly neighboring to the one used for MALDI imaging, while the Oil Red O-stained specimen is 16 tissue sections (corresponding to 192 μm) away. Histological images are representative of 2 biological replicates. Scale bars: a-e: 1 mm. **f** Penetration plots of BTZ-043 into both granulomas (Fig. 5e) present in the section. Black vertical lines indicate the edges of the granulomas. Gray areas inside the granulomas indicate the cellular regions, which mainly consist of foamy macrophages, and necrotic areas. Source data are provided as Source Data file and as Supplementary Data 2 and 8.

abundance reaches a plateau instead of immediately decreasing towards the center of the necrotic core as it does in the right granuloma.

## Sustained pulmonary BTZ-043 concentration above the MIC
For many anti-TB drugs, plasma concentrations are not predictive of concentrations in the infected lung[6]. We therefore investigated the

time course of BTZ-043 concentration in lung tissue harboring centrally necrotizing granulomas from Mtb-infected IL-13[tg] mice that have received 10 daily doses of BTZ-043 (250 mg/kg). At 0.5 h, 2 h, 4 h or 8 h after the last administration lesions were dissected from 2 mice per time point and cryosections were prepared for subsequent drug quantification by LC-MS/MS (Fig. 6a and Supplementary Fig. 8 for the respective biological replicates; Supplementary Data 3). BTZ-043

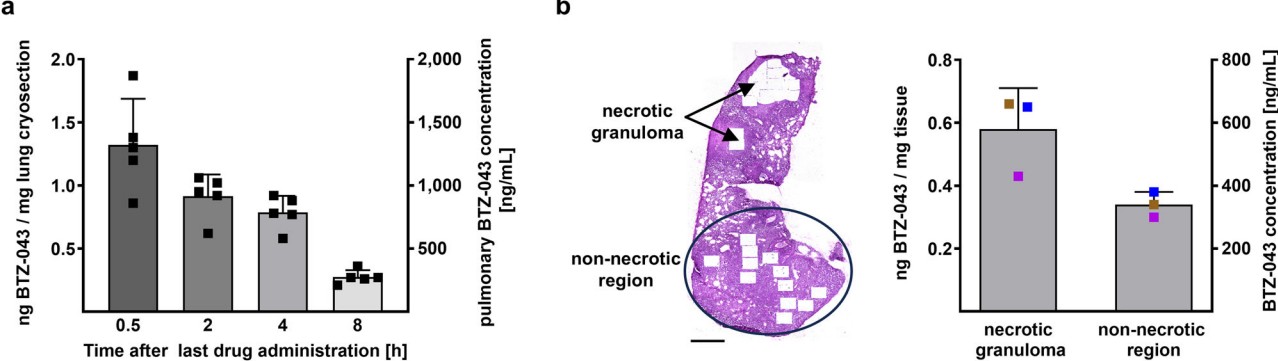

**Fig. 6 | Pulmonary BTZ-043 concentrations.** IL-13[tg] mice were infected with 263 CFU Mtb H37Rv and after 9 weeks, animals were treated with 250 mg/kg/day of BTZ-043 for 10 days prior to lung tissue collection at indicated time points after the last administration. **a** Change of pulmonary BTZ-043 concentration over time. Lung cryosections ($n = 5$) from 1 mouse per time point were prepared for BTZ-043 quantification by LC-MS/MS which is depicted as [ng/mg] on the left $y$ axis. The right $y$ axis shows the corresponding BTZ-043 concentrations [ng/mL] calculated based on an assumed tissue density of 1 g/mL. Data are shown as mean and SD. **b** BTZ-043 concentrations in necrotic granulomas and non-necrotic lung tissue 4 h after administration. Left panel: HE-stained lung section to visualize necrotic

granulomas and non-necrotic regions after sampling by LCM. Scale bar: 1 mm. Right panel: BTZ-043 concentrations determined in necrotic granulomas and non-necrotic regions by coupling LCM and LC-MS/MS which is depicted as [ng/mg] on the left $y$ axis. The right $y$ axis shows the corresponding BTZ-043 concentrations [ng/mL] calculated based on an assumed tissue density of 1 g/mL. Lung cryosections obtained from 1 mouse were used to collect 3 samples of each tissue type for subsequent LC-MS/MS measurements. Data represent mean and SD ($n = 3$; color code: data points with same color indicate LCM sampling from same cryosections). Source data are provided as Source Data file and as Supplementary Data 3 and 4.

concentrations between $1.12 \pm 0.19$ and $1.32 \pm 0.33$ ng/mg cryosection were already detected at 0.5 h after the last drug administration. Drug concentrations determined at 2 h after oral gavage ranged between $0.91 \pm 0.15$ and $1.10 \pm 0.25$ ng/mg cryosection, at 4 h between $0.79 \pm 0.12$ and $1.94 \pm 0.37$ ng/mg cryosection and at 8 h between $0.27 \pm 0.05$ and $0.65 \pm 0.06$ ng/mg cryosection. Importantly the lowest determined pulmonary concentration (0.27 ng/mg) measured 8 h after the last drug administration correlates to 270 ng/mL when assuming a tissue density of 1 g/mL. Since the in vitro MIC of BTZ-043 against H37Rv is 1 ng/mL[18], the pulmonary BTZ-043 concentrations reported in our study are substantially higher than the MIC at all time points investigated (in total $n = 8$ mice for data shown in Fig. 6a and Supplementary Fig. 8). This was further studied using LCM coupled with LC-MS/MS revealing a mean BTZ-043 concentration of 0.57 ng/mg in necrotic granulomas while the mean concentration in non-necrotic regions was 0.34 ng/mg (4 h after application) (Fig. 6b and Supplementary Fig. 9; Supplementary Data 4). This indicates an accumulation of BTZ-043 in the granuloma compared to the non-necrotic tissue.

### BTZ-043 retention in necrotic granulomas

To investigate the spatial distribution of BTZ-043 in more detail, lung sections from different time points after the last administration were investigated by MALDI imaging (Supplementary Data 2 and 8). To enable a credible comparison of the BTZ-043 distribution between post-dose time points, only sections containing granulomas with a similar histology (necrotic center surrounded by a cellular layer) were selected for this investigation. Post-measurement HE stainings showing the histology of the investigated sections for each time point are given in Fig. 7a–d. In these stainings, the necrotic center is visible in dark violet. The lighter violet area surrounding the necrotic center represents the cellular compartment mainly consisting of foamy macrophages. The BTZ-043 distribution 0.5 h and 2 h after the last administration was similar to the one shown in Fig. 5, with a higher intensity in the layer of foamy macrophages than in the surrounding tissue and a low intensity in the necrotic center (Fig. 7e–f). After 4 h, BTZ-043 was evenly distributed across the granuloma, including a full penetration of the necrotic center (Fig. 7g). Eight hours post-dose, BTZ-043 was still detected in the cellular compartment and the necrotic center of the lesion while the intensity in the surrounding tissue declined indicating a longer retention of BTZ-043 in centrally

necrotizing granulomas (Fig. 7h). Biological replicates for each time point are presented in the Supplementary Fig. 10. In total, we investigated 10 individual granulomas. In all of these samples a penetration of BTZ-043 into the lesion was observed. In addition, all granulomas obtained 4 h and 8 h after administration (in total $n = 4$ mice) showed a BTZ-043 accumulation in the necrotic lesions compared to the surrounding tissue.

### Discussion

Successful TB treatment depends on the ability of drugs to completely penetrate centrally necrotizing granulomas that harbor the majority of mycobacteria and their antimycobacterial activity within these lesions. Effective antibiotics like RIF, a drug that allowed shortening of treatment to 6 months and MXF, which was a necessary precondition of shortening to 4 months[43], showed at least some penetration into the cellular rim and necrotic center[5]. Importantly, efficient drug penetration into centrally necrotizing granulomas including all sublesional compartments is a prerequisite to avoid spatial or temporal monotherapy which may result in drug resistance. The development of new drugs and regimens for more effective and shorter therapy is often hampered by a knowledge gap between the late pre-clinical stage and phase I clinical studies. Consequently, to improve the predictive value of pre-clinical studies and prioritize the most promising compounds, drug candidates should be validated in advanced animal models that reflect the pathology of human TB with centrally necrotizing granulomas[44].

For a kinetic analysis of drug penetration into necrotic lesions, the optimal treatment dose has to be defined. For BTZ-043, so far only selected doses (37.5 mg/kg, 50 mg/kg or 300 mg/kg) had been analyzed in the BALB/c mouse model and treatment outcome was analyzed at a single time point revealing a significantly reduced bacterial load after 4 weeks of therapy[18,19]. In the present study the efficacy of BTZ-043 was comprehensively investigated during the chronic stage of TB infection by conducting a dose escalation study in Mtb-infected BALB/c mice and assessing the bacterial burden at different time points of treatment. This revealed not only a dose dependent but also a time dependent antimycobacterial activity of BTZ-043. Treatment with the 3 highest doses (250 – 1000 mg/kg/day) for 8 weeks efficiently reduced the bacterial load to the same extent, so that the dose of 250 mg/kg/day was identified as the LME for BTZ-043.

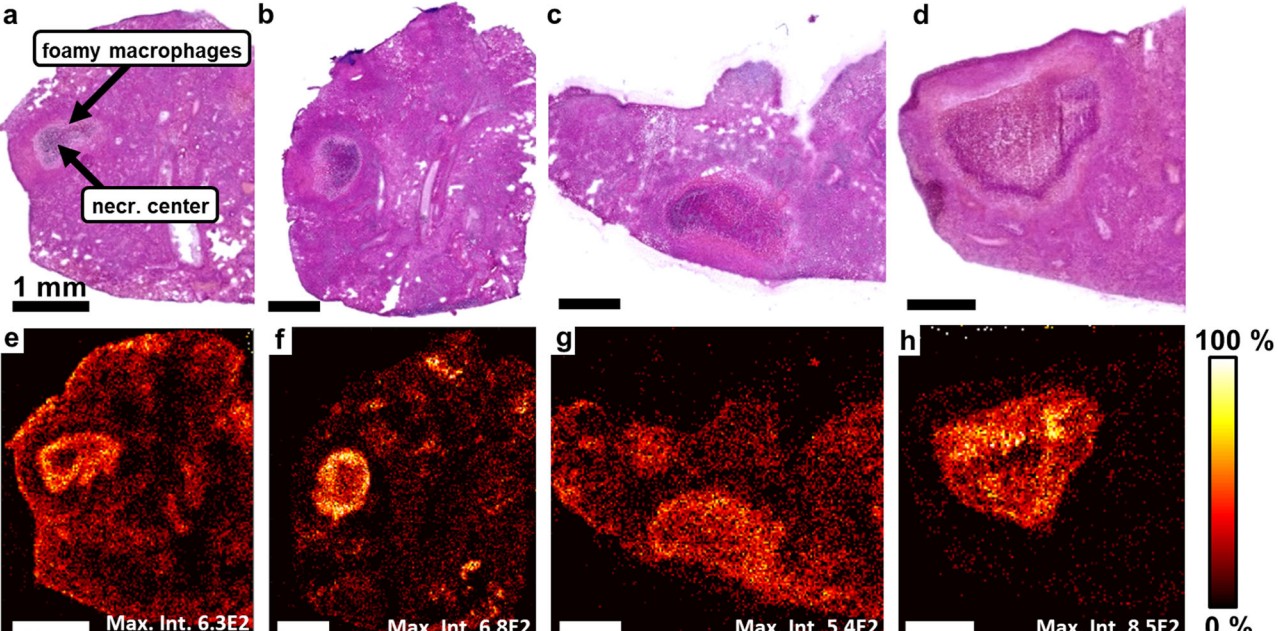

**Fig. 7 | Spatial and temporal distribution of BTZ-043 in centrally necrotizing granulomas of IL-13[tg] mice by MALDI imaging.** Mice were infected with 263 CFU Mtb H37Rv and treatment with 250 mg/kg/day of BTZ-043 for 10 days started 9 weeks after infection. Lung tissue was collected 0.5 h (a, e), 2 h (b, f), 4 h (c, g) and 8 h (d, h) after the last administration. **a**–**d** Post-measurement HE staining. **e**–**h** Distribution of BTZ-043 $[M + H]^+$ in these sections by MALDI imaging, pixel size 30 μm. Maximum intensities are given at the bottom of each image. Histological images are representative of 2 biological replicates. Scale bars: 1 mm. Source data are provided as Supplementary Data 2 and 8.

To determine the PK parameter most closely correlated with BTZ-043 efficacy we conducted a dose fractionation PK/PD study in BALB/c mice. BTZ-043$_{total}$ $C_{max}$ did not increase for daily doses above 50 mg/kg whereas higher doses resulted in a better antimycobacterial activity so that $C_{max}$ can be excluded as PK parameter driving efficacy. Divided daily dosing resulted in increased bactericidal activity compared to once daily dosing of the same total dose of 5 to 50 mg/kg/day. Since the BTZ-043$_{total}$ AUCs were linear over this dose range and plasma levels back to baseline prior to the second administration (7 – 8 h time difference), single or twice daily administration of the same total daily dose should result in the same AUC. This indicates that AUC is not responsible for the increased efficacy of the twice daily dosing. However, the fractionation of lower doses led to an increase in time with plasma levels above the MIC. Therefore, our results suggest that $T >$ MIC is the PK parameter most closely related to bactericidal activity of BTZ-043 in mice. For higher BTZ-043 doses, our PK study revealed a plateau for BTZ-043$_{total}$ $C_{max}$ and a non-linear increase for AUC. Therefore, our PK/PD data are in accordance with a recently published phase IIa study (early bactericidal activity (EBA) 250 mg – 1750 mg once daily for 14 days)[22] which also reports on a plateau in BTZ-043$_{total}$ exposures as well as no obvious further increase in bacterial killing above daily doses of 1000 mg BTZ-043 in TB patients.

While TB granulomas in humans are very heterogenous with a diverse spectrum of pathophysiological conditions, many lesions display necrosis in the center surrounded by a peripheral rim of fibrosis[45]. However, most wild-type mouse strains used for pre-clinical studies only develop cellular granulomas with little evidence of necrosis. Therefore, advanced mouse models replicating the pulmonary pathology of TB patients are of particular importance to optimize the predictive value of pre-clinical drug testing and to accelerate drug development. We have previously established the IL-13[tg] mouse model that reflects major pathophysiological conditions existing in human pulmonary TB lesions including highly stratified centrally necrotizing granulomas with a necrotic center that is surrounded by a zone of foamy macrophages and walled off from the adjoining tissue by

collagen encapsulation[10]. In contrast to other mouse models, infected IL-13[tg] mice also share pathogenetic and immunological mechanisms with human TB[46]. Importantly, we were able to show that drug distribution in necrotic lesions of IL-13[tg] mice reflects their penetration pattern in human TB lesions[12,13] so that the IL-13[tg] mouse model is ideal for the combination of drug distribution and efficacy studies. Intriguingly, treatment of IL-13[tg] mice with the determined LME for BTZ-043 for 10 days already reduced the pulmonary CFU counts significantly, compared to the bacterial burden of the pretreatment control group. BTZ-043 activity was also shown in 2 other mouse models developing necrotizing granulomas after Mtb infection[47,48]. While treatment of NOS2-deficient mice with 50 mg/kg/day BTZ-043 for two weeks hardly reduced the bacterial burden, a profound bactericidal activity was observed after 4 weeks of treatment[47]. Very recently Ramey et al.[48] reported a significant reduction of the pulmonary bacterial load in the C3HeB/FeJ substrain after 2 months of BTZ-043 therapy. In contrast to these studies, we provide evidence for EBA of BTZ-043, which is in accordance with its recently reported strong bactericidal activity in a phase IIa study in TB patients[22].

While the efficacy of BTZ-043 has been investigated in several mouse models[18,19,47,48] our study also addressed the emergence of drug resistance in vivo. BTZ-043 monotherapy neither resulted in the evolution of drug resistant mutants in the dose escalation study nor under conditions of centrally necrotizing granulomas. Furthermore, mutations in the *rv0678* gene which were recently reported to confer low-level resistance to the DprE1 inhibitors BTZ-043, PBTZ169, TBA-7371, and OPC-167832[30,31] were not selected during BTZ-043 monotherapy. This is most likely due to the high drug exposure achieved within the infected lung.

The BTZ-043-mediated potent reduction of pulmonary myco-bacterial counts is in line with its low in vitro MIC of 1 ng/mL against H37Rv[18]. The quantification of BTZ-043 in lung cryosections harboring centrally necrotizing granulomas in our study showed high pulmonary drug concentrations (270 – 1940 ng/g) until 8 h after administration. Coupling LCM with LC-MS/MS revealed a mean BTZ-

043 concentration of 570 ng/mL within TB granulomas 4 h after administration, which is at least two orders of magnitude above the in vitro MIC. Very recently the LCM LC-MS/MS approach was also employed for the quantification of BTZ-043 in lung compartments of Mtb infected C3HeB/FeJ mice[48]. While this study also demonstrated BTZ-043 penetration into necrotic granulomas, no statement can be made about the time course of the distribution or retention of BTZ-043, since, in contrast to our investigations, analyzes were only carried out at 1 h and 24 h post dosing.

As $T > MIC$ was found to be an important PK parameter for BTZ-043 efficacy, it can be assumed that the sustained pulmonary BTZ-043 concentration manifold above the in vitro MIC contributes to the substantial general and most importantly also lesional anti-mycobacterial activity observed in Mtb-infected IL-13[tg] mice in our study. Since the IL-13[tg] mouse model reflects the barrier of central necrotizing granulomas of TB patients, this lesional activity may have predictive value for the effect of BTZ-043 in humans.

MALDI imaging allows an accurate evaluation of drug distribution in tissue as this technique combines high resolution mapping of relative drug concentrations and histological information[49]. MALDI imaging was previously used to reveal a lesion and sublesion-specific distribution of first- and second-line drugs within TB granulomas[5,16,50]. Fluoroquinolones, a drug class which has allowed treatment shortening to 4 months, preferentially penetrate the cellular layers of necrotic granulomas with a poor diffusion into the acellular center[37,51], and bacteria in the necrotic core are not exposed to bactericidal concentrations of free drug in this compartment[52]. Also, the detection of bedaquiline by MALDI imaging revealed an accumulation within cellular regions surrounding granulomas relative to the necrotic core of lesions[11]. Likewise, MALDI imaging of the hydrophobic drug candidate SQ109, which has already completed phase II clinical trials with mixed results, detected strong signals in the cellular rim of centrally necrotizing lesions but only poor diffusion into the avascular caseum[53,54]. In summary, a penetration in the necrotic center has not been visualized for these antibiotics. However, a sufficient diffusion to the target bacteria in this compartment is essential for efficient treatment. In our study, the spatial and temporal distribution of BTZ-043 was assessed by high-resolution MALDI imaging within centrally necrotizing granulomas, as these harbor the majority of Mtb, with extracellular bacteria in the necrotic core and intracellular bacteria in the zone of foamy macrophages[10,42,55]. Early after administration an increased BTZ-043 intensity was detected in cellular areas of centrally necrotizing granulomas and the surrounding lung tissue. High resolution mapping and detailed penetration analysis of BTZ-043 at 2 h post-treatment in correlation with comprehensive histological characterization of lesions revealed a high abundance in the zone of lipid-rich foamy macrophages implying an accumulation of the lipophilic BTZ-043[18] within these immune cells. Importantly, at later time points after administration BTZ-043 penetrates the entire granuloma including the necrotic core. The drug distribution at different time points also revealed a prolonged retention of BTZ-043 in the cellular and acellular compartments of granulomas as illustrated in Supplementary Fig. 11. Taken together, the early BTZ-043 accumulation within the rim of foamy macrophages suggests that these lipid-rich immune cells act as reservoir for BTZ-043. The subsequent penetration into the necrotic core is likely due to a slower passive diffusion of the released drug which occurs in the absence of vascularization or active transport systems[56]. Such a passive diffusion is supported by our permeability assay which demonstrates that BTZ-043 is a permeable drug. In any case, BTZ-043 is a clinical-stage drug for which an accumulation and efficient penetration of centrally necrotizing TB granulomas has been shown by MALDI imaging. Consequently, BTZ-043 reaches not only intracellular mycobacteria within the zone of foamy macrophages but also extracellular bacilli in the necrotic core which is often the most problematic niche in terms of drug penetration.

First evidence for a local antimycobacterial activity of BTZ-043 within necrotic granulomas comes from our careful histological comparison of AFB in lesions of untreated and treated IL-13[tg] mice which revealed a prominent reduction of AFB in the rim of foamy macrophages of BTZ-043 treated mice. In addition, our quantitative analysis of Mtb 16S rRNA expression in necrotic areas of granulomas isolated by LCM revealed that BTZ-043 acts very efficiently against Mtb in this compartment and reduces the bacterial burden by 2 $\log_{10}$. Consequently, our study demonstrates that BTZ-043 not only penetrates all compartments of TB granulomas but also exerts its anti-mycobacterial activity against intracellular mycobacteria within the zone of foamy macrophages as well as extracellular bacilli in the necrotic core of TB granulomas.

In summary, our study visualizes an efficient penetration and accumulation of a clinical-stage TB drug in centrally necrotizing granulomas and demonstrates its lesional activity. In addition, the concentration of BTZ-043 in these necrotic lesions was shown to be manifold above the MIC resulting in a potent lesional activity and general efficacy of BTZ-043 in Mtb-infected IL-13[tg] mice. Tuberculosis regimens with at least two drugs having activity in necrotic lesions are likely required to substantially shorten the treatment of patients and prevent the emergence of resistance[57]. Based on the penetration, local concentration, and lesional activity of BTZ-043 in centrally necrotizing granulomas, our study provides an important indication for BTZ-043's ability to complement existing antibiotics. Therefore BTZ-043 could contribute to future treatment shortening regimens, prevent relapse, and thwart the development of drug resistance. The medical scientists of the present study started the clinical evaluation of several BTZ-043-based regimens in 2023 within UNITE4TB[24] and in 2024 within the PanACEA consortia.

## Methods

### Animal ethics statement

Animal experimentation conducted at the Research Center Borstel (RCB) was in accordance with the German regulations of the Society for Laboratory Animal Science (GV-SOLAS) and the European Health Law of the Federation of Laboratory Animal Science Associations (FELASA). All animal experiments were approved by the animal research ethics committee of the federal state of Schleswig-Holstein prior to permission by the Ministry of Energy, Agriculture, the Environment, Nature, and Digitalization (Kiel, Germany; permits 3-1/15, 69-6/16, and 84-9/20). For experiments performed at Johns Hopkins University, all procedures involving the care and use of animals in the study were reviewed and approved by the Johns Hopkins University Animal Care and Use Committee (protocol #MO15M479). The care and use of animals were conducted in accordance with the principles outlined in the guidance of the Association for Assessment and Accreditation of Laboratory Animal Care (AAALAC), the Animal Welfare Act, the American Veterinary Medical Association (AVMA) Euthanasia Panel on Euthanasia, and the Institute for Laboratory Animal Research (ILAR) Guide to the Care and Use of Laboratory Animals.

### Mice

Female BALB/c mice were purchased from JANVIER LABS (Le Genest Saint-Isle, France) or Charles River Laboratories (Sulzfeld, Germany, or Wilmington, MA, USA). IL-13[tg] mice expressing murine *il13* under the control of the human CD2 locus control region[58] [Tg(CD4-Il13) 431Anjm] backcrossed to the BALB/c genetic background were bred under specific-pathogen-free conditions at the Christian-Albrechts-University of Kiel (Kiel, Germany) or the Max-Planck-Institute for Evolutionary Anthropology (Leipzig, Germany). Mtb-infected mice were maintained in individually ventilated cages (IVC, Ebeco, Castrop-Rauxel, Germany) under BSL-3 conditions at the RCB. Female BALB/c mice receiving drug treatment without prior infection were housed under standard conditions at the RCB.

## BTZ-043 dose escalation in Mtb-infected BALB/c mice

Aerogenic infection of female BALB/c mice with an intermediate dose of Mtb H37Rv was performed in a Glas-Col inhalation exposure system. For aerosolization, frozen bacterial aliquots with a known titer were thawed, single cell suspension prepared, diluted in sterile water and loaded into the inhalation exposure nebulizer unit[59]. Five mice were sacrificed the following day to determine the number of bacteria implanted in the lung and 3 weeks later to determine the pulmonary bacterial burden at the start of treatment (n = 5 mice). BTZ-043 was formulated every 3 weeks by suspending BTZ-043 (microcrystalline, containing 1.8% Tween 80) in 1% CMC sodium salt and Tween 80 added to a final concentration of 0.5%. BTZ-043 suspensions were stored at 4 °C and, prior to use, acclimated to room temperature and mixed to achieve homogenous suspensions before administration. INH was dissolved in distilled water, stored at 4 °C and prepared weekly. Treatment started 3 weeks after infection and drugs were administered by oral gavage 5 days per week with daily doses of 50, 100, 250, 500 or 1000 mg/kg body weight for BTZ-043, 25 mg/kg body weight for INH or CMC/Tween 80 as vehicle control. After 4, 6 and 8 weeks of treatment 5 to 6 mice per therapy group were sacrificed, lungs were aseptically removed, weighed, and homogenized. Tenfold serial dilutions of lung homogenates were plated on Middlebrook 7H10 agar supplemented with 10% bovine serum and incubated at 37 °C for 3 weeks. Colonies on plates were enumerated and results expressed as $\log_{10}$ CFU per lung for the comparison of treatment efficacy.

## BTZ-043 dose-fractionation in Mtb-infected BALB/c mice

Female BALB/c mice, 8-10 weeks old, were aerosol-infected using a Glas-Col inhalation exposure system and a thawed aliquot of broth culture having a known bacterial titer 3 weeks prior to treatment. Mice were sacrificed for lung CFU counts the following day (n = 3 mice) and 3 weeks later at the initiation of treatment (n = 3 mice) to determine the number of CFU implanted and the number present at the start of treatment, respectively. Both, BTZ-043 microcrystalline (containing 1.8% Tween 80) and BTZ-043 amorphous (hot melt extrudate BTZ-043 / Soluplus 1:5) particles were suspended in 1% CMC sodium salt and Tween 80 added to a final concentration of 0.5%. On weekly basis, INH was dissolved in distilled water and kept refrigerated at 4 °C. The untreated control group received no treatment. The BTZ-043 formulations were mixed by shaking before administration. Treatment started 3 weeks after infection. Drugs were administered 5 days per week, by gavage, either once or twice daily. For all BID dosing, the time between administrations was 7 – 8 h. The treatment scheme is presented in the Supplementary Table 3. After 6 weeks of treatment, 3 to 4 mice per treatment group were sacrificed, lungs were aseptically removed and homogenized in glass grinders. Serial 10-fold dilutions of the homogenates were prepared, and 0.5 mL aliquots were plated on 7H11 agar plates[60]. Baseline lung $\log_{10}$ CFU counts were assessed the day after aerosol infection (day -20) and at treatment initiation (day 0). Treatment efficacy was assessed by comparing lung $\log_{10}$ CFU counts after 6 weeks of treatment.

## PK investigation

Naïve BALB/c mice were dosed orally for 5 days with BTZ-043 at doses used in the BTZ-043 dose-fractionation study in Mtb-infected BALB/c mice (2.5, 5, 50, 250 mg/kg/day) to reach steady state (Study no. 832.220.5501). On day 5, plasma samples were collected for bioanalytical analysis pre-dose and at 0.5 h, 1 h, 2 h, 4 h and 8 h post-dosing (see Supplementary Table 1 for more details).

BTZ-043 partially forms the unstable metabolite M2 in vivo, a hydride Meisenheimer complex[61]. This metabolite can only be quantified indirectly by reverse conversion to the parent compound. The amount of BTZ-043 measured after reverse conversion of M2 is called BTZ-043$_{total}$. In an amber 0.5 mL Eppendorf vial, 20 µL of plasma were

mixed with 2 µL of methanol (MeOH). 10 µL of formic acid (FA; Suprapur, Merck, Darmstadt, Germany) in MilliQ Water (50% (v/v)) were added and the samples mixed by vortexing. Afterward, the samples were mixed for additional 60 min at 40 °C in a thermoshaker. Protein precipitation at room temperature followed with 60 µL acetonitrile (ACN)/internal standard. Samples were mixed by vortexing and after centrifugation (9727 x g / 10 min / room temperature) supernatants were transferred into an amber autoinjector vial (0.4 mL). LC was performed on an Agilent 1100 Series HPLC (Agilent Technologies, Santa Clara, CA, USA) using a Inertsil ODS-4 C$_{18}$ column (GL Sciences Inc. 2.1 × 75 mm, 3 µm particle size) with an Inertsil ODS-4 guard column (1.5 × 10 mm, 3 µm particle size) at a column temperature of 25 °C. Mobile phase solvent A: 0.1% FA. Solvent B: MeOH with 0.1% FA. The gradient started at 50% B and was increased to 75% until minute 1 which was held until minute 3. B was raised to 100% at 3.1 minutes and held until minute 6. At 6.1 min B was dropped to 50% and held until minute 10 resulting in a total runtime of 10 minutes. Injection volume was 1 µL at high concentration (20 – 7000 ng/mL) and 2 µL at low concentration levels (5 – 2000 ng/mL). Autosampler temperature was set at 5 °C. Electrospray ionization (ESI) multiple reaction monitoring (MRM) MS/MS on an API 2000 (Applied Biosystems, Waltham, MA, USA) triple quadrupole mass spectrometer was used for signal detection. The transition 432.1 → 82.9 in positive ion mode was used for quantification. Spray voltage was 4.5 kV and source temperature 400 °C. Data analysis was performed in Analyst 1.4.2.

## Determination of drug permeability and prediction of drug absorption in Caco-2 monolayers

An in vitro permeability assay with Caco-2 cells was performed according to a standard protocol[34]. Briefly, Caco-2 cells (DSMZ no.: ACC 169) were grown on permeable inserts (Corning® 12-well Transwell with 0.4 µm pore polyester membrane insert (diameter of 12 mm) from Merck, Darmstadt, Germany) for 3 weeks to obtain a monolayer of polarized Caco-2 cells. The apical and basolateral compartment contained a volume of 0.4 mL and 1.2 mL, respectively. The permeation of BTZ-043 and PZA was monitored from apical to the basolateral compartment and vice versa by taking samples from the receiver compartment after different time points (0, 10, 30, 60, 90 and 120 min). Atenolol (reference compound for low permeability) and propranolol (reference compound for high permeability) were used as controls. All experiments were performed in triplicates for each transport condition. Prior and subsequent to the incubation period, each insert was checked for integrity and tightness of the Caco-2 monolayer by measuring the transepithelial electrical resistance (TEER) with Epithelial Volt/Ohm Meter 3 (EVOM3, Word Precision Instrument, Sarasota, FL, USA). Furthermore, the integrity of the Caco-2 monolayer after to the permeation experiment was checked through addition of Lucifer Yellow to the apical compartment and incubation for 1 h. The extent of permeation (%) was determined by fluorescence measurement using a SPARK platereader (Tecan, Grödig, Austria). The subsequent quantification of drugs is described in the Supplementary Information section.

## BTZ-043 treatment of Mtb-infected IL-13$^{tg}$ mice

Aerogenic infection of IL-13$^{tg}$ mice (male and female) with an intermediate dose of Mtb H37Rv using a Glas-Col inhalation exposure system was performed as described above. Five mice were sacrificed the following day to determine the number of bacteria implanted in the lung. After 9 weeks of infection the development of centrally necrotizing granulomas was histologically confirmed, and the pulmonary bacterial burden was determined (n = 4 mice). Mice received 10 daily doses of 250 mg/kg BTZ-043 by oral gavage to reach steady state conditions and were euthanized at defined time points (0.5 h, 2 h, 4 h, 8 h) after the last administration. Lungs were aseptically removed and tissue containing macroscopically visible lesions was snap frozen in

liquid nitrogen for subsequent cryosectioning. To assess the bacterial burden, the remaining lung tissue was homogenized, tenfold serial dilutions of organ homogenates were plated on Middlebrook 7H10 agar supplemented with 10% bovine serum and incubated at 37 °C for 3 weeks. Colonies on plates were enumerated and results expressed as $\log_{10}$ CFU per lung for the comparison with baseline lung $\log_{10}$ CFU counts determined before the start of treatment. To assess the anti-mycobacterial activity of BTZ-043 over a prolonged period, IL-13[tg] mice were infected via the aerosol route with an intermediate dose of Mtb H37Rv (341 CFU). Ten weeks after infection, mice were treated 5 times a week with 250 mg/kg of BTZ-043 or the vehicle control and the pulmonary bacterial burden was determined after 2 weeks and between 4 and 6 weeks of treatment. Some mice receiving the vehicle suspension became moribund so that the second analysis time point comprises a treatment period of 4 to 6 weeks.

### Analysis of BTZ-043 resistance

The development of BTZ-043 resistance in Mtb infected BALB/c or IL-13[tg] mice was investigated by drug susceptibility testing using the agar proportion method[62]. Briefly, Mtb containing lung homogenates were prepared and cultivated from BTZ-043 treated BALB/c mice (50, 100, 250, 500 or 1000 mg/kg/day for 6 and 8 weeks), vehicle control mice, and from BTZ-043 treated IL-13[tg] mice (250 mg/kg/day for 10 days). Drug susceptibility testing was performed by plating tenfold serial dilutions of Mtb cultures onto Middlebrook 7H10 agar supplemented with 10% OADC and containing 1, 5 or 10 ng/mL BTZ-043 or DMSO. Bacterial growth was determined after 4 weeks of incubation at 37 °C. Additionally H37Rv served as control. In our study, development of drug resistance was defined based on growth on solid medium supplemented with 5 or 10 ng/mL BTZ-043. Samples for DNA sequencing were obtained by preparing lung homogenates from representative mice that were treated with the 3 highest BTZ-043 doses (250, 500 or 1000 mg/kg/day for 6 and 8 weeks) and untreated mice. Mycobacteria Growth Indicator Tubes were inoculated with the lung homogenates and incubated at 37 °C. Mtb DNA was isolated from the resulting bacterial cultures. The DNA sequence of the entire *mmpR5* gene (*rv0678*; forward primer 5'-CGG AAC CAA AGA AAG TGC GG-3', reverse primer 5'-TTG CGA GGT TGC TCA TCA GT-3') and the region of the *dprE1* gene that includes codon Cys387 (*rv3790*; forward primer: 5'-AGC AAT TGC CTG CGA AAC TG-3'; reverse primer 5'-GGG GTT TCC TAC GGC ATC AA-3') was determined by Sanger sequencing (ABI 3500DX, Life Technologies, Darmstadt, Germany) using the the BigDye™ Terminator v1.1 Kit (Thermo Fisher Scientific Baltics UAB, Vilnius, Lithuania). The primers were designed to correspond to sequences with GenBank accession number AL123456. Collection of data was conducted using the 3500 Series Data Collection Software 3; Build id 3500 v.3.3 and data analysis using Sequencing Analysis Software 7.0 Build id B01.

### Histology and immunohistochemistry

Cryosections were fixed with 4% paraformaldehyde/PBS (24 h or 1 h for non-irradiated or irradiated sections, respectively) prior to staining with haematoxylin and eosin (HE), ZN or Heidenhain's azan trichrome. For immunohistochemical analysis ZN stained cryosections were probed with rabbit polyclonal CD68 antibody (1:200, ab125212, Abcam, Cambridge, UK) which was detected with biotinylated goat anti-rabbit antibody (1:100, 111-065-144, Jackson Immunoresearch, Suffolk, UK)[13]. To visualize lipid droplets, irradiated cryosections were fixed (10% ice-cold formalin/PBS), washed with 60% 2-propanol, and incubated in Oil Red O solution (O0625, Sigma-Aldrich, Taufkirchen, Germany). After washing (60% 2-propanol and water), Gill's haematoxylin was used for counterstaining. Images were acquired on a BX41 light microscope (Olympus, Hamburg, Germany) using cellSens imaging software (Olympus, Hamburg, Germany) or NIS-Elements software (Nikon, Badhoevedorp, The Netherlands). Post MALDI imaging

HE staining was performed with the following protocol[40]: the matrix layer was washed off with MeOH. Rehydrated (2 min in 100%, 70%, 40% EtOH then 100% $H_2O$) samples were submerged in Mayers haematoxylin solution for 12 min followed by 10 min submersion in $Na_2CO_3$ (1%) and rinsed with distilled water. Counterstaining was performed by 2-min submersion in 0.5% acidified eosin Y.

### Tissue cryosectioning and irradiation

Serial cryosections (12 μm) were cut from unembedded lung tissue at −25 °C using a Leica CM1850 or CM3050s cryostat (Leica Microsystems, Wetzlar, Germany) and thaw mounted onto adhesive glass slides (SuperFrost™ Langenbrinck, Emmendingen, Germany) as described by Treu et al.[40]: a small area of the pre-cooled glass slide was warmed from behind by finger contact to ensure proper sample mounting. Sections were stored at −80 °C until further processing. Cryosections were γ-irradiated (5.85 kGy, BIOBEAM 8000, Gamma-Service Medical GmbH, Leipzig, Germany) on dry ice for the inactivation of Mtb[13].

### LCM and subsequent quantification of Mtb 16S rRNA expression by MBLA

LCM was applied using a PALM Microbeam (Zeiss, Oberkochen, Germany) to isolate necrotic areas of granulomas from snap-frozen methanol-fixed histological sections of lungs from treated IL-13[tg] mice for further analysis. Two infection experiments were carried out, in which treatment was given for 2 weeks and for 2 and 4 – 6 weeks, respectively. To determine the bacterial load in areas of central granuloma necrosis, ~4 – 5 million μm² of 12 μm thick sections per mouse were collected by LCM. Areas of necrotic lesions from IL-13[tg] mice treated for 2 and 4 - 6 weeks were transferred by LCM in RLT buffer (Qiagen, Hilden, Germany) containing 1 % beta-mercaptoethanol (Sigma-Aldrich, Taufkirchen, Germany) and frozen. In duplicate reactions, RNA was extracted (RNeasy Plus Micro Kit, Qiagen, Hilden, Germany) and the expression of Mtb 16S rRNA was measured in a reverse transcriptase quantitative PCR using the QuantiTect Multiplex RT-PCR NoRox kit (Qiagen, Hilden, Germany) performed in the first experiment on a Rotorgene Q (Qiagen, Hilden, Germany) and in the second experiment on a LightCycler 480 II (Roche, Basel, Switzerland). Primers (16S rRNA forward, 5'-GTG ATC TGC CCT GCA CTT C-3'; 16S rRNA reverse, 5'-ATC CCA CAC CGC TAA AGC G-3') and probe ([FAM] AGG ACC ACG GGA TGC ATG TCT TGT[BHQ1])[33] were purchased from Eurofins Genomics (Ebersberg, Germany). To generate a standard curve, 16S rRNA was isolated in parallel from culture-derived bacteria of known concentrations (taken from the Vitalbacteria kit®, St. Andrews, UK, or kindly provided by Marit Neumann, RCB, respectively). To assess the individual performance of RNA extraction, an internal control (taken from the Vitalbacteria kit® or kindly provided by Isobella Honeyborne, University College London, respectively) was added to each reaction, the amplification of which was measured after reverse transcription in the 2nd fluorescence channel. The bacterial load was automatically calculated (Rotor-Gene Q-Rex or LightCycler 480 software version LCS480 1.5.1.62, respectively) based on the automatically generated standard curve of the co-extracted 16S rRNA from culture-derived bacteria of known concentrations. Results were normalized against the volume of tissue area, which was microdissected using LCM.

### LC-MS/MS of tissue sections

Extraction procedure for lung cryosections: Prior to sample preparation for LC-MS/MS measurements, the weight of cryosections was determined based on the area and thickness of sections and by assuming a density of 1 g/cm³. Cryosections were detached from the glass slides using sterile water[13]. After drying of lung cryosections in a SpeedVac, samples were reconstituted with 25 μL LC-MS-grade water, 200 μL ACN (reserpine concentration: 12.5 ng/mL) and 25 μL 1% FA.

Samples were vortexed and afterward centrifuged for 10 min at $15,000 \times g$, room temperature. ~200 µL of the resulting supernatant were transferred in a 1.5 mL Eppendorf tube and re-centrifuged under the same conditions. <u>Quantification by LC-MS/MS:</u> Liquid chromatography was performed on an Agilent 1100 Series HPLC (Agilent Technologies, Santa Clara, CA, USA) using a SeQuant® ZIC®-HILIC column (Merck Millipore SeQuant, 2.1 inner diameter x 150 mm length with 5 µm particle size, pore-size 200 Å) at a column temperature of 30 °C. The mobile phase consisted of 1% FA (solvent A) and ACN (solvent B). The gradient started at 90% B at a flowrate of 0.5 mL/min. After 1 minute of isocratic conditions, the percentage of ACN was decreased to 2% B until minute 4. At minute 4, the flowrate was increased to 0.8 mL/min. The gradient was kept isocratic at 2% B with a flowrate of 0.8 mL/min for 6 minutes until minute 10. Afterward, the percentage of ACN was re-increased to 90% B until minute 15 and the flowrate re-decreased to 0.5 mL/min after minute 19. These conditions were maintained for 1 minute, so that the total run time was 20 minutes. The autosampler temperature was set to 4 °C and sample injection volume was 5 µL. The Waters Micromass Quattro Premier XE triple quadrupole mass spectrometer (Waters Corporation, Milford, MA, USA) using ESI was operated in positive ion mode using MRM. The transition $432.1 \rightarrow 292.3$ of BTZ-043 (cone voltage 30 eV, collision energy 30 eV, dwell time 0.1 s) was used as the quantifier. The reserpine transition $608.6 \rightarrow 194.9$ (cone voltage 30 eV, collision energy 35 eV, dwell time 0.1 s) was used as the internal standard. The cone gas- and desolvation gas flow were set to 100 L/h and 800 L/h, respectively. The extractor voltage was 3.0 V. We optimized the capillary voltage, the source-, and the desolvation gas temperature and chose 3.0 kV, 90 °C, and 450 °C, respectively. MassLynx 4.1 and TargetLynx (Waters Corporation, Milford, MA, USA) were used for operating the platform and quantifying the samples, respectively.

### BTZ-043 calibration curve for quantification

Lung homogenate prepared from naïve mice was dried in a SpeedVac and reconstituted in 100 µL LC-MS-grade water. Then 800 µL ACN, containing reserpine (12.5 ng/mL per sample) as reference for quantification and ionization, and 100 µL 1% FA were added. The solution was incubated for 10 min at room temperature with continuous shaking at 1300 rpm. To avoid floating particles, the solution was centrifuged for 10 min at $15,000 \times g$. The resulting supernatant was collected in a separate tube and was once more centrifuged under the same conditions (10 min, $15,000 \times g$). Calibration BTZ-043 standard was prepared by diluting a stock solution in extracted lung tissue homogenate with a curve range as 0.001–0.25 µg/mL.

### LCM coupled LC-MS/MS

Necrotic granulomas and non-necrotic regions were dissected from 2 to 3 γ−irradiated serial lung cryosections (thickness: 12 µm) using a PALM Microbeam (Zeiss, Oberkochen, Germany). Laser-dissected tissues were captured in an ascorbic acid solution (2.5 mg/mL) to prevent the degradation of BTZ-043 and stored at -80 °C. Samples were thawed on ice before drying in the SpeedVac. For reconstitution 80 µL of an 80% ACN/ 20% FA, containing 20 ng/mL reserpine and $d_4$-BTZ-043, were added. The samples were vortexed for 1 min and kept in an ultrasonic bath for 3 min. Subsequently, the samples were centrifuged (10 min, $15,000 \times g$). The supernatant was transferred into a glass vial and 5 µL was injected into the LC. The LC settings and MS parameters were the same as described for the tissue mimetics (Supplementary Methods). As internal standard $d_4$-BTZ-043 was utilized, with the $m/z$-transitions $436.0 \rightarrow 86.0$ (cone voltage of 30 V, collision energy of 30 eV, dwell time of 0.2 s) as quantifier. With the expected very low concentrations of the analyte in the LCM material, the deuterated standard was applied to better mimic matrix effects and recovery for this analysis. For calibration curve preparation lung homogenate from drug-naïve mice was added to 80% ACN/20% FA (1%), containing

reserpine (as QC) and $d_4$-BTZ-043 (20 ng/mL per sample) to the final solution containing about 0.44 µg tissue/mL, which equals the average weight of the obtained material of the LCM. After centrifugation (5 min, $15,000 \times g$) the supernatant was spiked with BTZ-043 to yield a calibration curve ranging from 0.00015–0.025 µg/mL.

### MALDI imaging and penetration analysis

Lung cryosections were shipped on dry ice from the RCB to the University of Bayreuth for MALDI imaging analysis and stored at −80 °C. Sections were brought to room temperature inside a desiccator for 15 min. DCTB matrix (5 mg/mL, 1:1 chloroform/ethanol acidified with 0.1 Vol.% trifluoroacetic acid) was applied using a pneumatic sprayer system built in house. Deuterated standard BTZ-043 D4 was applied (0.5 µg/mL in 1:1 acetone/water total applied volume 25 µL) using a pneumatic sprayer built in house. Imaging measurements were performed using an AP-SMALDI 10 (TransMIT GmbH, Gießen, Germany) atmospheric pressure MALDI imaging source equipped with a $\lambda = 337$ nm N$_2$ laser operating at repetition rate of 60 Hz coupled to a Q Exactive HF (Thermo Fisher Scientific, Bremen, Germany) orbital trapping mass spectrometer. For MS data acquisition the Thermo software Q Exactive HF tune Version 2.9 was used and the MALDI source was controlled by the provided TransMIT software Master Control Program Version 3.9. Measurements were conducted with 30 laser shots per pixel and a mass range of $m/z$ 420-540 with a step size of $10 \times 10$ µm or $30 \times 30$ µm. MS spectra were analyzed using the proprietary software Xcalibur 4.0 from Thermo Fisher Scientific. Ion images and RGB overlays were generated in MSiReader Version 1.0[63] with a bin width of 2 ppm after conversion of the Thermo RAW files to the imzML format (imzML Converter Version 2.0.4). In order to assess possible artifacts as a result of different tissues properties in granuloma and surrounding tissue, an isotopically labeled BTZ-043 standard was applied in an additional application step. This experiment showed that the detected BTZ-043 signal did not depend on the tissue properties and therefore ion suppression effects can be ruled out as a source of relevant artifacts (see Supplementary Fig. 12 for more details). Penetration plots were generated using our in-house penetration analysis tool[12]. The generation of the granuloma edge was based on the co-detected ion $m/z$ 482.36050 highlighting the granuloma areas. To determine the on tissue region, the co-detected ion $m/z$ 488.44620 was used. See the Supplementary Fig. 7 for single ion images. All measurements were conducted using the matrix ion $m/z$ 539.25715 [2 M + K]$^+$ as the reference for internal mass calibration[40]. Mass accuracies across imaging datasets are given as the root mean square error (RMSE) of the $m/z$ values in ppm of all pixels containing the targeted ion within a ± 3 ppm window of the theoretical $m/z$. The Supplementary Table 4 gives the RMSE values of BTZ-043 in measurements included in this study. MS/MS experiments were conducted with a precursor isolation window of ± 0.2 $m/z$ using higher energy collision induced dissociation (HCD 26).

### Statistical analysis

If applicable, statistical analysis was performed using Prism 9 (Graphpad Software, San Diego, CA, USA). Quantifiable data are expressed as the means of individual determinations and standard deviations (SD). Log$_{10}$-transformed CFU data were evaluated by two-tailed Student's $t$ test, by a one-way or two-way ANOVA followed by Bonferroni's post hoc test for multiple comparisons.

### Reporting summary

Further information on research design is available in the Nature Portfolio Reporting Summary linked to this article.

## Data availability

All data necessary to evaluate the conclusions in the study are provided in the paper and/or the Supplementary Materials. HPLC-MS/MS

data of BTZ-043 in tissue have been deposited at Zendo (https://doi.org/10.5281/zenodo.14179430). MS imaging data is available at METASPACE. Source data are provided with this paper.

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

## Acknowledgements

The authors are very grateful for funding by the German Federal Ministry of Education (BMBF) via the German Center of Infection Research (DZIF) within the "Thematic Translational Unit Tuberculosis" (TTU TB, grant number 02.705 (CH, NH, MH), 02.709 (DS), 02.710 (CH, NH, MH), 02.806 (AR, CH, NH, MH), 02.810 (AR, CH, KW, NH, MH), 02.813 (DS), 02.814 (AR, CH, KW, NH, MH), 02.901 (CH, MH), 02.902 (AR, CH, NH, MH), TI 07.003 (MG)), BMBF InfectControl 03TTA01 (FK, LuG), Leibniz Center for Photonics in Infection Research (LPI, contract number 13N15718) (SG), the German Research Foundation (DFG) INST 91/3731-FUGG (AR) and – SFB 1357 – 391977956 (AR) and the TechnologieAllianzOberfranken (TAO) (AR). Hapila GmbH, Gera produces BTZ-043 and provides all analytical standards for KUM and HKI. We would like to thank Matthias Ochs, Margarita Neun and Elena Roman-Paucar for support in MALDI sample preparation. We are grateful to Ilka Monath, Christine Keller, Sarah Vieten and Gerhard Schultheiß for organizing the animal facilities at the RCB and Kiel. We would like to thank the National Reference Center for Mycobacteria at the RCB for MGIT testing and Fenja Boysen and Doreen Beyer for technical support in performing drug susceptibility testing. Our thanks go to Isobella Honeyborne for providing the IC (University College London) and Marit Neumann for bacteria to generate the 16S rRNA standard. We are grateful to Nikolas Jakobs for his support in performing IHC staining. We thank Simone Thomsen for technical support in performing LC-MS/MS measurements. We are grateful to Ute Möllmann for sharing her expertise regarding the preparation of BTZ-043 formulations. We thank Oliver Aehlig and Valerie Kerndl for their technical support in performing the CaCo-2 assay.

## Author contributions

A.R., C.H., and K.W. conceived the study. A.R. and K.W. coordinated the work. A.T. and K.W. wrote an early draft of the manuscript, and extensive data and discussion have been added subsequently. A.R., C.H., and K.W. edited the manuscript, and comments were provided by all authors. A.T., J.K.-H., and A.R. developed, analyzed, and interpreted the MALDI imaging analysis. E.S. and Le.G. performed additional MALDI imaging experiments. C.H. and K.W. analyzed and interpreted animal experiments at RCB. N.A., F.M., and D.S. acquired, analyzed, and interpreted the LC-MS/MS analysis. D.H. acquired, analyzed, and interpreted DNA sequencing data and substantially contributed to the design of drug susceptibility testing. M.G. developed and performed the coupling of LCM with MBLA. J.V. conducted LCM, A.H. subsequently isolated RNA and accomplished MBLA. M.G. and C.H. interpreted the lesional activity. M.H. and F.K. initiated and coordinate BTZ-043 development. J.D., N.H., and M.H. provided BTZ-043 and PK data and contributed to the conceptualization of mouse studies. A.-K.L., J.V., and A.H. conducted animal experiments at RCB. P.J.C., S.T., and E.L.N. conducted, analyzed, and interpreted the dose-fractionation study at JHU. S.G., Lu.G., and F.K. acquired, analyzed, and interpreted the Caco-assay. All authors read and approved the manuscript. A.T., J.K.-H., and F.M.: second authors who contributed equally to this work.

## Funding

## Competing interests
NH has received a grant for conducting a study on delpazolid, a TB drug candidate, from LegoChem biosciences, to his institution. ELN received research funding from TB Alliance, Gates Medical Research Institute, and Janssen to his institution. MH, NH, and JD are employees of LMU Klinikum the institution that is developing BTZ-043 and have received funding from the German government. All other authors declare no competing interests.

## Additional information

[1]Bioanalytical Sciences and Food Analysis, University of Bayreuth, Bayreuth, Germany. [2]Thematic Translational Unit Tuberculosis, German Center for Infection Research (DZIF), Partner Site Munich-Bayreuth, Munich, Germany. [3]Division of Bioanalytical Chemistry, Research Center Borstel, Leibniz Lung Center, Borstel, Germany. [4]Thematic Translational Unit Tuberculosis, German Center for Infection Research (DZIF), Partner Site Hamburg-Lübeck-Borstel-Riems, Hamburg, Germany. [5]Institute of Infectious Diseases and Tropical Medicine, LMU University Hospital, LMU Munich, Munich, Germany. [6]National and WHO Supranational Reference Center for Mycobacteria, Research Center Borstel, Borstel, Germany. [7]Division of Infection Immunology, Research Center Borstel, Leibniz Lung Center, Borstel, Germany. [8]Center for Tuberculosis Research, Division of Infectious Diseases, Johns Hopkins University School of Medicine, Baltimore, Maryland, USA. [9]Transfer Group Antiinfectives, Leibniz Institute for Natural Product Research and Infection Biology, Leibniz-HKI, Jena, Germany. [10]Robotic-assisted Discovery of Antiinfectives, Leibniz Institute for Natural Product Research and Infection Biology, Leibniz-HKI, Jena, Germany. [11]Fraunhofer Institute for Translational Medicine and Pharmacology ITMP; Immunology, Infection and Pandemic Research, Munich, Germany. [12]German Center for Lung Research (DZL), Airway Research Center North (ARCN), Research Center Borstel, Leibniz Lung Center, Borstel, Germany. [13]Unit Global Health, Helmholtz Zentrum München, German Research Center for Environmental Health (HMGU), Neuherberg, Germany. [14]These authors contributed equally: Axel Treu, Julia Kokesch-Himmelreich, Franziska Marwitz. ✉e-mail: Andreas.Roempp@uni-bayreuth.de; kwalter@fz-borstel.de

