## [Peer Review File · Nature Communications]

The clinical-stage drug BTZ-043 accumulates in murine tuberculosis lesions and efficiently acts against *Mycobacterium tuberculosis*

Corresponding Author: Dr Kerstin Walter

Version 0:

Reviewer comments:

Reviewer #1

(Remarks to the Author)

The manuscript of Treu and colleagues describes the preclinical evaluation of BTZ-043, a new inhibitor of *M. tuberculosis* currently under clinical development. The authors used mass spectrometry imaging (MSI) analysis to demonstrate that the BTZ-043 can accumulate within the necrotising environment of granulomas in a murine infection model. These results are important as ever as new drug to treat TB and drug resistant TB are urgently required and BTZ-043 based on the current trials and preclinical evaluation seems to be a good candidate. The authors add to their previous work on using MSI on the IL-13tg mouse model that better mimics disease pathology by forming necrotising granulomas. They present robust methodology and analysis of their data and conclusions are supported by their findings. Overall, the authors findings are important and relevant for TB drug discovery as well as in the field of MSI as new you can now quantify localised areas with lung tissue with the amount of drug present. However, my overall concern is how translatable this is in humans and is there enough evidence to support the IL-13tg mouse model as a predictive tool of human outcomes and disease, especially considering possible differences in drug metabolising enzymes between the two organisms. Specific comments are below:

The article title is a bit general and as the findings were conducted in a murine model it would be appropriate to indicate this on the title of this manuscript.

Line 32-33 and 58: Authors mention that "Centrally necrotizing granulomas that harbor *Mycobacterium tuberculosis* (Mtb) are the hallmark of human tuberculosis (TB)." I think the granuloma as a structure, in general, is considered the hallmark of TB disease.

Line 140: "As the efficacy did not significantly increase beyond 250 mg/kg/day" – are there any data to suggest evolution of resistance against the BTZ-043? Any possibly toxicity at high concentrations?

In the in vitro permeability experiments using the Caco-2 assay, the authors used PZA as a positive control; however, the permeability coefficient calculated for BTZ-043 is considerably less than the one of PZA. It would have been better to include in addition to PZA another drug that is less permeable for a better comparison, especially when suggesting that BTZ-043 has a similar potential to penetrate these lesions as does PZA.

Line 235: The authors suggest that 10 days of treatment with BTZ-043 reduced the bacterial load significantly by 0.9 logs. What is the statistical test done here to determine significance here?

In Fig. 3: it seems that the granuloma look different between untreated group and the group treated with BTZ-043. Is there any explanation of these differences?

Fig. 4: The authors declare that BTZ-043 accumulates inside necrotising granulomas; however, from this figure there are differences in the presence of the drug inside granulomas. The authors did refer to the heterogeneity of granulomas present but it is not clear to me how many granulomas were tested using MSI to conclude the reasoning that this new drug indeed accumulates inside granulomas. What about other markers that are present in specific granulomas with high BTZ-043 accumulation. Is there another mechanism behind this based on cells and the environmental conditions being present

there?

Lines 326, 327: Since there is about 200x higher than MIC concentration of BTZ-043 in the lung, what about data showing the generation of resistance against BTZ-043? Especially as Figure 1 shows that there is limited reduction of bacterial load after 8 weeks of treatment? Have the authors looked at this and if not are there any data on drug resistance against BTZ-043 from murine evaluations?

Line 341-342: How did the authors correct for bias on their selection of granulomas for BTZ-043 retention experiments? In total how many different granulomas were investigated?

Reviewer #2

(Remarks to the Author)

The study reports the first visualisation of penetration, accumulation, and retention of a clinical TB drug in centrally necrotising granulomas.

The work represents a significant advance on previous MSI studies in this field e.g. Prideaux et al 2011, 2015.

All data acquired appears to be valid as are the interpretation tools and methodologies employed.

The methods sections are well written and the work could be reproduced from them.

Reviewer #3

(Remarks to the Author)

Summary

This study focuses on the detection and accumulation of the antimycobacterial drug, BTZ-043, in human-like TB lesions taken from the IL-13tg mouse model. BTZ-043 is a DprE1 inhibitor that has demonstrated strong preclinical bactericidal activity against drug-sensitive and extensively drug resistant clinical strains of Mtb and is currently being evaluated in phase II clinical trials. For the most part, the study is methodical, and the manuscript is clearly articulated.

While BTZ-043 has proven efficacy against multiple clinical isolates of Mtb, its penetration into mature caseous necrotic lesions and subsequent quantification in correlation to its known MIC remains to be reported. This information is important as it can aid in the development of optimal dosing regimens for effective Mtb sterilization. The study reports the 'first' visualization and accumulation of a clinical-stage TB drug in human-like centrally necrotizing TB lesions, which is a misleading statement, used a lot throughout the manuscript. Mass spectrometry imaging of antimycobacterial drugs is not novel and has been used routinely by the Dartois lab for both clinical-stage and clinical approved drugs for many years now (for example, PMID: 34228540, PMID: 30427309, PMID: 29618565, PMID: 26185484, PMID: 27227164, PMID: 34252307). There have also been a number of preclinical efficacy studies in both chronic and human-like TB mouse models (PMID: 19299584, PMID: 28821804). This, along with the lack of patient sample correlates or novel mechanistic/pharmacodynamics findings, limits the impact of the study as presented. This study is of value as it reports data for shorter dosing schedule (10 days) and does provide drug penetration information, but it is more suitable for a target PK audience, and is therefore not recommended for publication in Nature Communication.

There are also concerns regarding the interpretation of the analytical data.

General Comments

p. 13 line 237-238: BTZ-043 treatment reduced bacterial burden by 86% does not match the data presented for the log₁₀ CFU reduction of 0.9 presented following 10 days of therapy. How was this % reduction calculated and from where?

- BTZ-043 bactericidal activity has previously been demonstrated in cellular and human-like necrotizing lesions during a study using NOS2-deficient mice. This study warrants discussion in the context of the results presented herein.

p. 15 line 269-272: Evaluation of drug variability would be more suitably carried out using known spiked drug concentrations in tissue mimetics as this would reduce the biological variability observed and provide absolute quantification data on the impact of gamma IR on drug stability. Degradation is structure-dependent for IR and can vary from 0-50+%.

p.16 line 298-300: The left granuloma is described as containing a thicker macrophage layer when this appears to be a mostly cellular or an early necrotic lesion, with what minimal necrosis. Whereas the right lesion has a larger necrotic core and appears to be a more mature necrotic lesion. A significant difference in drug accumulation and mean intensity is observed between these lesions due to the difference in necrosis. These pathologies are important for drug penetration and efficacy as penetration into cellular lesions is more efficient and Mtb is readily sterilized in these lesions. This sentence could use additional description of granuloma pathology and type in relation to drug penetration.

p. 16 Fig. 4e-f and description: There is a lot of focus on sublesional penetration but little attempt has been made to analytically characterize the accumulation of BTZ-043 in the cellular vs necrotic regions. Mean intensity is given for the entire inside of the caseum for two lesions with very different pathologies and the data reflects this. It would be beneficial to mark where the cellular macrophage regions end and the necrotic regions begin for the data presented.

p. 17 from line 317, Fig 5 and Supplemental Table 3: The data presented in the manuscript and in supplemental table 3 do not correlate. For example, the mean concentrations presented in the supplementary for “a biological replicate” are: 0.5 h = 1.12 ± 0.19 ; 2 h = 1.10 ± 0.25 ; 4h = 1.94 ± 0.37 ; 8h = 0.65 ± 0.06 ng/mg cryosections. The values given in the text and figure are as follows: 0.5 h = 1.32 ± 0.33 ; 2 h = 0.91 ± 0.15 ; 4h = 0.79 ± 0.12 ; 8h = 0.27 ± 0.05 ng/mg cryosections.

The description and methodology do not indicate how many mice were used to generate the cryosection LC-MS data. It appears to be one sample for the figure and one for the supplemental with five technical replicates (5 cryosections from the same sample). Information on biological and technical replicates should be included to ensure experimental and analytical rigor.

There is also no data or information on BTZ-043 plasma levels from these time-points or if normalization to plasma concentration was carried out prior to calculating lung concentrations.

The concentrations of BTZ-043 are reported to be substantially higher than the MIC at all time-points investigated. This is not supported by the data shown. This data is for cryosections of lung tissue and therefore reflects normal, cellular and necrotic regions and does not provide information on the concentration of BTZ-043 within the necrotic core to correlate with MIC.

p. 19 Fig 6 and description: It is unclear if these images are processed on the same intensity scale across the time course study. This information or the addition of the absolute intensity values for each ion image would enable a more accurate analysis of the accumulation of BTZ-043 at each time post-dose.

Specific comments

p. 6 line 80-81: The description of the requirement of a spatial resolution in the range of 10 μm for reliable identification of drug compounds in sub-organ structures is misleading. The average macrophage diameter is $\sim 21 \mu\text{m}$ in diameter, foamy macrophages are larger. Additionally, drug accumulation is rarely cell-specific and more regional-specific. A resolution of 50 μm is sufficient to detect drug accumulation in the center of necrotic regions as well as differential accumulation in the cellular regions of granulomas. As has been shown repeatedly by the Dartois lab (for example, PMID: 30427309).

p. 15 line 268-269: γ -irradiation is not the only sterilization technique for infected tissue. Whole tissue heat sterilization has been shown for a number of pathogen causing microbes that require biocontainment (PMID: 25966989). More recently, an on-slide heat-sterilization technique was reported for MSI studies of drugs, lipids and metabolites from Mtb infected lung tissue (PMID: 34672552).

Version 1:

Reviewer comments:

Reviewer #1

(Remarks to the Author)

The revised manuscript of Römpp and colleagues addressed the initial concerns and recommendations. The work is original and important in the field of drug development for tuberculosis and this version is considerably improved.

One minor comment on the line 227-8 of the merged manuscript, the authors claimed that “By 2 weeks of treatment, this expression was already diminished in the BTZ-043 treated animals compared to the vehicle group (Fig. 4b, right panel).” Looking at that panel it doesn’t seem that it is diminished but rather reduced. I would choose a different verb here.

Reviewer #3

(Remarks to the Author)

The revision by Römpp and colleagues of the manuscript entitled “The clinical-stage drug BTZ-043 accumulates in murine tuberculosis lesions and efficiently acts against Mycobacterium tuberculosis” has significantly improved the manuscript. The authors carried out extensive additional studies and have further clarified methodological parameters such as sample numbers, CFU calculations and statistics. I commend the authors for their hard work in advancing the impact of their manuscript. The addition of actual drug quant, lesional drug penetration calculations for granulomas with slightly differing pathologies, and an initial evaluation of drug resistance, has all contributed to the improvement of their manuscript. This manuscript is important to help inform on the further development of optimal dosing of BTZ-043 during clinical trial development.

I do have a few comments:

Figure 6 and corresponding discussion: Showing data for n=1 biological replicate is not analytically sound. Technical replicates of adjacent cryosections will of course have similar drug concentrations. Statistical tests are usually carried out on the mean of the technical replicates for n=5-10 biological replicates. This is important for population variability.

The argument the authors present for the requirement of 10 μm or high-resolution imaging based on the size of the granuloma is quite weak. A 30-50 μm pixel resolution of even the smallest granulomas would provide enough information to determine if the drug was able to reach the core of the necrotic center. This is also shown by the authors in their 30 μm pixel images for the spatial and temporal distribution of BTZ-043 in Figure 7. At or near single cell imaging is only really required

in situations where differentiation of neighboring single cells with different populations or phenotypes are required. Or when there truly is a heterogenous accumulation of drug in small cellular regions, etc..

Dear reviewers,

we would like to sincerely thank you for your very knowledgeable and helpful comments which have helped us to further improve our manuscript. We have carefully revised our manuscript accordingly. This includes substantial additional experiments:

- Laser-capture-microdissection (LCM) combined with LC-MS/MS to quantify local drug concentrations in selected lung tissue.
- LCM combined with Mtb 16S rRNA quantification to determine lesional activity.
- Drug susceptibility testing to evaluate resistance.
- Tissue mimetics to address gamma-irradiation effects.

The implementation of these experiments makes it necessary to acknowledge the contribution of 5 new co-authors. Following a shift in author contributions (e.g. Axel Treu has left the university), we have also decided to rearrange the author list in the revised version: Andreas Römpp is now the first author while Kerstin Walter remains last author.

Our detailed response for each comment is given below (**line numbers refer to the pdf document with tracked changes**).

Yours faithfully,

Kerstin Walter and Andreas Römpp

PS: In order to be fully compliant with the *Nature Communications* guidelines (in particular the "Reporting Summary" questionnaire), we have amended and updated a number of statistical details in the revised version. This also included a recalculation of all mass accuracy values (RMSE) in Supplementary Table 5.

Reviewer #1:

The manuscript of Treu and colleagues describes the preclinical evaluation of BTZ-043, a new inhibitor of *M. tuberculosis* currently under clinical development. The authors used mass spectrometry imaging (MSI) analysis to demonstrate that the BTZ-043 can accumulate within the necrotising environment of granulomas in a murine infection model. These results are important as ever as new drug to treat TB and drug resistant TB are urgently required and BTZ-043 based on the current trials and preclinical evaluation seems to be a good candidate. The authors add to their previous work on using MSI on the IL-13tg mouse model that better mimics disease pathology by forming necrotising granulomas. They present robust methodology and analysis of their data and conclusions are supported by their findings. Overall, the authors findings are important and relevant for TB drug discovery as well as in the field of MSI as now you can now quantify localised areas with lung tissue with the amount of drug present. However, my overall concern is how translatable this is in humans and is there enough evidence to support the IL-13tg mouse model as a predictive tool of human outcomes and disease, especially considering possible differences in drug metabolising enzymes between the two organisms. Specific comments are below:

Response

We thank the reviewer for these very helpful comments. We hope that, above all, the additional infection experiments with the subsequent determination of the local concentration and local antimycobacterial activity of BTZ-043 in the center of necrotic granulomas will better express the relevance of our study, especially at the translational level.

In tuberculosis (TB), the barrier of the centrally necrotizing granulomas is a major reason for the need for combination therapy over 6 months (in contrast to other lung infections). Animal models that form human-like centrally necrotizing granulomas after infection with *Mycobacterium tuberculosis* (Mtb) (C3HeB/FeJ mice, rabbits) have been described to represent ideal translatable tools to analyze the penetration of antibiotics into the center of the lesions. We demonstrated previously that our IL-13^{tg} mouse model recapitulates the penetration of antibiotics into human centrally necrotizing granulomas (Walter 2022, Kokesch-Himmelreich 2022). Our present study corroborates findings of other granuloma necrosis models that BTZ-043 treatment reduces the bacterial load in lungs (PMIDs 28821804, 36975792, 37791784) which again proves the comparability of IL-13^{tg} mice to these.

Manuscript modification:

Lines 513-519: We now take a closer look at Ramey's publication in particular and compare the effect of BTZ-043 treatment in the different mouse models.

With regard to the assumed translational value of these granuloma necrosis animal models, two aspects are important which cannot be modelled by standard animal models.

(1) Most importantly, the EBA of BTZ-043 in IL-13^{tg} mice was also observed in a phase IIa clinical trial in which the medical scientists of the present study were involved (Heinrich 2023, DOI: 10.2139/ssrn.4601314) which is another proof for the translatability of our study. In TB patients, regimens with at least two drugs having activity in necrotic lesions are likely

required to substantially shorten treatment to less than 6 months of the standard therapy (Savic, Union conference 2023). Because the IL-13^{tg} mouse model confirms the penetration into centrally necrotizing granulomas of antibiotics, the distribution of which is known in human lesions (Walter 2022, Kokesch-Himmelreich 2022) IL-13^{tg} mice are in general a valid predictive tool. Our study now additionally demonstrates that BTZ-043 penetrates all lesional compartments including the necrotic core, which is a niche that is difficult for antibiotics to reach. Therefore, BTZ-043 has the potential to complement other anti-TB drugs in novel regimens.

Manuscript modification:

Lines 519-521: We mention the phase IIa study and the comparable EBA in IL-13^{tg} mice and TB patients.

Abstract and lines 615-626: We now state more clearly the requirement that the penetration properties of the individual drugs must be known for new regimens and that the IL-13^{tg} mouse model is able to provide information about these properties - especially for BTZ-043.

(2) In TB, the pharmacodynamic (PD) properties of drugs are above all important in the centre of necrotic granulomas, i.e. locally. PD data measured by the concentrations in plasma and the general antimycobacterial activity can only provide limited information about the actual efficacy of a drug in patients. With our animal model, we can not only determine the penetration of new antibiotics like BTZ-043 into the lesions. We are additionally making a significant contribution to translation to inform on the relationship of concentration and activity of BTZ-043 within the lesions with an efficient local efficacy and local concentrations exceeding the MIC many times over – a property that is relevant for TB but cannot be visualised by determining the concentration of BTZ-043 in plasma or in whole lung homogenates. In this context, it may be useful to emphasise, that the authors of this study not only carry out animal experiments to evaluate new antibiotics, but the medical scientists of the present manuscript are also responsible for clinical studies on the effect of BTZ-043 in humans. Hence, our mouse data will influence the design of further clinical trials, as they are considered important factors for shortening therapy. In addition, these data are even considered important for submission to ethics and regulatory authorities.

Manuscript modification:

Lines 269-278: Description of results for local antimycobacterial activity of BTZ-043.

Lines 395-400: Description of results for lesional concentration of BTZ-043.

Lines 532-539: Discussion of lesional concentration in context of published literature.

Lines 596-602: Discussion of local antimycobacterial activity.

New Figure 4: Data for local antimycobacterial activity.

New Figure 6b: Data for lesional concentration.

We also briefly summarized these new results in the abstract and discussed these results also in relation to the possible translational information.

By MALDI imaging we could convincingly demonstrate the particular distribution and retention of BTZ-043 in lesions. However, while the caseum activity assay of BTZ-043 (to predict penetration and activity *in vitro*) in the studies presented by Ramey (PMID 37791784) and Robertson (PMID 34370580), was negative for all tested DprE-1 inhibitors, we have some evidence to believe that for a covalently binding drug as BTZ-043 such a short-term assay might underestimate the potential of a drug that irreversibly binds to the target enzyme for a long period of time (unpublished data showing that protein-bound BTZ-043 can still be detected in a human ADME study after months). As mentioned above, further evidence is given in our revised manuscript that BTZ-043 is in fact active in the lesions *in vivo*. Quantitative real time PCR of Mtb 16S rRNA expression in the necrotic tissue of granulomas isolated by laser-capture-microdissection revealed a strong reduction of the bacterial load corroborating our finding that BTZ-043 accumulates in the lesions of IL-13^{tg} mice. Because the local exposure of BTZ-043 is more than sufficient to effectively reduce the local bacterial load and we observed a saturation of BTZ-043 at high dosing concentrations not only in mice but also in humans (Heinrich 2023, DOI: 10.2139/ssrn.4601314), we assume that possible differences in drug metabolism have no effect on the predictive power of our model.

We hope that with this complex answer and the additional experiments, we can now convincingly demonstrate that the IL-13^{tg} mouse model is of translational relevance.

The article title is a bit general and as the findings were conducted in a murine model it would be appropriate to indicate this on the title of this manuscript.

Response

We thank the reviewer for this comment and must admit that the title is indeed too general. We have supplemented the text with the mention "murine".

Manuscript modification:

Lines 1-2: “The clinical-stage drug BTZ-043 accumulates in murine tuberculosis lesions and efficiently acts against *Mycobacterium tuberculosis*”

Line 32-33 and 58: Authors mention that “Centrally necrotizing granulomas that harbor *Mycobacterium tuberculosis* (Mtb) are the hallmark of human tuberculosis (TB).” I think the granuloma as a structure, in general, is considered the hallmark of TB disease.

Response

In fact, different formulations are used in the field (e.g. „A hallmark of human tuberculosis (TB) infection is the development of granulomatous lesions with central caseous necrosis“; in Gautam et al. Am J Respir Cell Mol Biol 52: 708). Since active TB depends on central necrotizing granulomas (as well as the subsequent caseation and cavity formation) which is usually present at the start of therapy this status is important for our research question.

Manuscript modification:

Lines 51-52: We would like to suggest the following formulation: „**The development of granulomas with central necrosis** harboring *Mycobacterium tuberculosis* (Mtb) **is** the hallmark of human tuberculosis (TB). “

Line 140: “As the efficacy did not significantly increase beyond 250 mg/kg/day” – are there any data to suggest evolution of resistance against the BTZ-043? Any possibly toxicity at high concentrations?

Response

We thank the reviewer for drawing our attention to the evolution of bacterial drug resistance which is a major concern in the field of drug discovery and development. We therefore thoroughly assessed the development of BTZ-043 resistance in our dose escalation study which is described in the results section (**lines 185-196**) the discussion (**lines 522-528**) and methods (**lines 772-797**) of the revised manuscript. We did not observe the evolution of drug resistance during BTZ-043 treatment. It is furthermore noteworthy that shortly after the introduction of bedaquiline, which is the core drug for the current treatment of MDR TB, Mtb strains with bedaquiline resistance were identified and variants in the gene *Rv0678* are the most current resistance mechanism against bedaquiline. Since *Rv0678* mutations were also reported to confer low-level resistance to the DprE1 inhibitors TBA-7371, OPC-167832, PBTZ169, and BTZ-043 (PMIDs: 35920665, 36786606) we additionally sequenced the *Rv0678* gene from a mixture of Mtb that were isolated from representative BTZ-043 treated mice. In all samples sequenced we did not detect a mutation in the *Rv0678* gene. The relevant information for these additional analyses are included in the respective sections of the revised manuscript.

Regarding possible BTZ-043 toxicities at high treatment doses, we have thoroughly monitored the health status of drug treated mice and did not observe any indication that could point to adverse side effects. At necropsy, we detected enlarged stomachs in mice that were treated BTZ-043 and this was most pronounced in mice receiving the highest dose for 8 weeks. We interpreted this observation as a tendency of decelerated digestion or reduced peristaltic activity of the stomach at high BTZ-043 doses and prolonged treatment.

Our PK study showed for higher BTZ-043 doses a plateau for C_{max} and non-linear increase for AUC in plasma. We therefore assumed that a saturation of intestinal uptake is reached at about 250 mg/kg daily BTZ-043 dosing which is the reason for the efficacy plateau observed in the dose escalation study at higher doses. Since we did not investigate the intestinal uptake in further detail, we did not mention this aspect in the manuscript.

Manuscript modification:

Lines 185-196: Description of drug susceptibility results.

Lines 522-528: Discussion of results in context of published literature.

Lines 772-797: Description of methods for the analysis of BTZ-043 resistance and *Rv0678* sequencing.

In the in vitro permeability experiments using the Caco-2 assay, the authors used PZA as a positive control; however, the permeability coefficient calculated for BTZ-043 is considerably less than that of PZA. It would have been better to include in addition to PZA another drug that is less permeable for a better comparison, especially when suggesting that BTZ-043 has a similar potential to penetrate these lesions as does PZA.

Response

We thank the reviewer for bringing this point to our attention. For better comparison with our determined permeability coefficient of BTZ-043, we have included permeability data from the literature for additional anti-TB drugs which are less permeable than PZA. We now include fluoroquinolones and linezolid which have permeability values in the same range as BTZ ($8-18 \cdot 10^{-6}$ cm/s) but show different penetration behaviour. We conclude that MALDI imaging is necessary in order to confirm and assess the distribution behaviour of anti-TB drugs.

Manuscript modification:

Lines 280-312: the Caco-2 assay paragraph was moved and is now positioned before the MALDI imaging results

Lines 291-312: comparison to other drugs has been adjusted accordingly.

Supplementary Figure 2: in order to streamline the text, experimental data for PZA has been moved to the Supplementary information

Line 235: The authors suggest that 10 days of treatment with BTZ-043 reduced the bacterial load significantly by 0.9 logs. What is the statistical test done here to determine significance here?

Response

For data shown in Figure 3, statistical analysis was carried out using a two tailed unpaired student's *t* test on \log_{10} -transformed CFU values which has been clarified in the figure caption of the revised version of the manuscript.

Manuscript modification:

Lines 1198-1200: “Data represent mean and SD ($n=4$ or 7) and statistical analysis was performed by an unpaired student’s t test (two-tailed) on \log_{10} -transformed CFU data (** $p=0.0071$). One experiment representative of 2 is shown.”

In Fig. 3: it seems that the granuloma looks different between untreated group and the group treated with BTZ-043. Is there any explanation of these differences?

Response

IL-13^{tg} mice develop a broad spectrum of granulomas after Mtb infection ranging from highly stratified centrally necrotizing granulomas over less well stratified necrotic granulomas to loosely organized cellular granulomas. Even though treatment of IL-13^{tg} mice with BTZ-043 for 10 days had a substantial impact on acid fast bacilli in macrophages, our microscopic analyses did not reveal effects on the overall composition, structure, or size of granulomas. For instance, highly stratified necrotic granulomas of similar size were observed in untreated IL-13^{tg} mice (depicted in Figure 3b (left panel) but also in BTZ-043 treated animals (depicted in Figure 5 a-d which was formerly Figure 4 a-d). We clarified this aspect in the revised manuscript.

Manuscript modification:

Lines 264-265: “A change in granuloma size or composition due to BTZ-043 treatment was not observed.”

Fig. 4: The authors declare that BTZ-043 accumulates inside necrotising granulomas; however, from this figure there are differences in the presence of the drug inside granulomas. The authors did refer to the heterogeneity of granulomas present, but it is not clear to me how many granulomas were tested using MSI to conclude the reasoning that this new drug indeed accumulates inside granulomas. What about other markers that are present in specific granulomas with high BTZ-043 accumulation. Is there another mechanism behind this based on cells and the environmental conditions being present there?

Response

A closer inspection of the two granulomas in Figure 5 (formerly Figure 4) revealed that the different structure is mainly due to different regions which have been sectioned. This also explains differences in drug distribution. We have modified the text in order to explain this situation.

Manuscript modification:

Lines 355-360: “The right granuloma is sectioned in its central region showing a prominent necrotic core. The left granuloma is sectioned in a more peripheral region so that the cellular layer which mainly consists of

foamy macrophages is more prominent. A neutrophil-rich zone between the macrophage layer and the necrotic core is present in both granulomas.”

In total we investigated ten individual granulomas with MSI - and they all showed the same general pattern:

- 1) Penetration of BTZ-043 was detected in ALL measured granulomas.
- 2) ALL granulomas analyzed early after dosing (0.5 h and 2 h) showed elevated levels in the rim of lipid-rich foamy macrophages.
- 3) An accumulation (compared to the surrounding tissue) was observed for all granulomas obtained 4 h and 8 h after drug administration.

This means that differences in penetration we observed were primarily due to time after drug administration. Based on our observations we assume that the lipophilic drug BTZ-043 first accumulates in the lipid rich rim of foamy macrophages which serves as a reservoir. The dimensions of this cellular area depend on the region of the granuloma which has been sectioned (as discussed above). Subsequently BTZ-043 slowly penetrates the necrotic center which is also a lipid rich environment. This aspect was already covered in the discussion of the original manuscript (now lines 574-585).

Manuscript modification:

Lines 422-425: We have extended our original statement of granulomas “showing a comparable penetration behavior” by the following text. “In total, we investigated 10 individual granulomas. In all of these samples a penetration of BTZ-043 into the lesion was observed. In addition, all granulomas obtained after 4 and 8 h BTZ-043 showed an accumulation in the necrotic center compared to the surrounding tissue.”

Lines 326, 327: Since there is about 200x higher than MIC concentration of BTZ-043 in the lung, what about data showing the generation of resistance against BTZ-043? Especially as Figure 1 shows that there is limited reduction of bacterial load after 8 weeks of treatment? Have the authors looked at this and if not are there any data on drug resistance against BTZ-043 from murine evaluations?

Response

The potential evolution of drug resistance is of major importance. We have therefore investigated the development of BTZ-043 resistance in drug treated IL-13^{tg} mice. We did not observe the evolution of drug resistance under conditions of centrally necrotizing granulomas. While the efficacy of BTZ-043 has been shown in a few mouse studies so far, to our knowledge, none of these studies has evaluated the development of BTZ-043 resistance. Thanks to the reviewer’s comments, our study is the first to address this important issue *in vivo* using the mouse model of TB.

Manuscript modification:

Lines 250-252: “The generation of resistance to BTZ-043 was not observed as Mtb from drug treated IL-13^{tg} mice did not grow on agar plates containing 5 or 10 ng/mL BTZ-043.”

Lines 522-528: Discussion of results in the context of published literature.

Lines 772-782: Description of methods for the analysis of BTZ-043 resistance.

Line 341-342: How did the authors correct for bias on their selection of granulomas for BTZ-043 retention experiments? In total how many different granulomas were investigated?

Response

Lung tissue was sectioned and H&E staining was performed at regular intervals to check for the presence of granulomas and to evaluate their histological structure. Granulomas that showed a clear stratification, i.e. a necrotic center surrounded by a cellular area, were chosen for BTZ-043 retention experiments with MALDI imaging. In total, 10 different granulomas were investigated by MALDI imaging.

Manuscript modification:

Lines 408-409: “To enable a credible comparison of the BTZ-043 distribution between post-dose time points, only sections containing granulomas with a similar histology (necrotic center surrounded by a cellular layer) were selected for this investigation.”

Line 422: “**In total, we investigated 10 individual granulomas.**”

Reviewer #2:

The study reports the first visualisation of penetration, accumulation, and retention of a clinical TB drug in centrally necrotising granulomas.

The work represents a significant advance on previous MSI studies in this field e.g. Prideaux et al 2011, 2015.

All data acquired appears to be valid as are the interpretation tools and methodologies employed.

The methods sections are well written and the work could be reproduced from them.

Response

We highly appreciate the reviewer's awareness about the significant contribution of our work to the field of MALDI imaging and anti-TB drug development.

Reviewer #3:

Summary

This study focuses on the detection and accumulation of the antimycobacterial drug, BTZ-043, in human-like TB lesions taken from the IL-13tg mouse model. BTZ-043 is a DprE1 inhibitor that has demonstrated strong preclinical bactericidal activity against drug-sensitive and extensively drug resistant clinical strains of Mtb and is currently being evaluated in phase II clinical trials. For the most part, the study is methodical, and the manuscript is clearly articulated.

While BTZ-043 has proven efficacy against multiple clinical isolates of Mtb, its penetration into mature caseous necrotic lesions and subsequent quantification in correlation to its known MIC remains to be reported. This information is important as it can aid in the development of optimal dosing regimens for effective Mtb sterilization. The study reports the ‘first’ visualization and accumulation of a clinical-stage TB drug in human-like centrally necrotizing TB lesions, which is a misleading statement, used a lot throughout the manuscript. Mass spectrometry imaging of antimycobacterial drugs is not novel and has been used routinely by the Dartois lab for both clinical-stage and clinical approved drugs for many years now (for example, PMID: 34228540, PMID: 30427309, PMID: 29618565, PMID: 26185484, PMID: 27227164, PMID: 34252307). There have also been a number of preclinical efficacy studies in both chronic and human-like TB mouse models (PMID: 19299584, PMID: 28821804).

This, along with the lack of patient sample correlates or novel mechanistic/pharmacodynamics findings, limits the impact of the study as presented. This study is of value as it reports data for shorter dosing schedule (10 days) and does provide drug penetration information, but it is more suitable for a target PK audience, and is therefore not recommended for publication in Nature Communication.

Response

We agree with the reviewer that MALDI imaging of anti-TB drugs is not new – and we never claimed that. We also fully agree that the Dartois lab has pioneered this approach, and we now mention their contribution specifically in the introduction of our revised manuscript (**lines 108-109**). We had cited all but 2 of the mentioned publications in our original manuscript. We have now included these two (PMID: 34252307 und PMID: 29618565) and also the latest study from the Dartois lab (PMID: 38581700) in the revised manuscript.

Concerning our statement, we state that we *report the “first visualization of an efficient penetration, accumulation, and retention of a clinical stage TB drug into centrally necrotizing granulomas”*. The publications cited by the reviewer either report on compounds that have already been approved (PMID: 30427309, PMID: 29618565, PMID: 26185484, PMID: 27227164, PMID: 34252307) or they report on clinical stage compounds for which visualization by MALDI imaging of an efficient penetration, accumulation and retention in centrally necrotizing granulomas has not been shown (PMID: 34228540). This is an important difference in the context that our data can be used in the design of current clinical studies as outlined below.

Concerning the formulation of being the “first”, we agree that we might have been a bit too enthusiastic in stressing it repeatedly. We have removed this specification in two cases in the main text: introduction (**line 149**) and discussion (**line 587**).

We also agree with the reviewer that some preclinical BTZ-043 efficacy studies have been published before and we have already discussed these two studies in our originally submitted manuscript. However, the study conducted in BALB/c mice (PMID: 19299584) investigated only two different BTZ-043 doses (37.5 mg/kg/day and 300 mg/kg/day). NOS2-deficient mice, which also develop a human like TB pathology, received daily BTZ-043 doses of 50 mg/kg/day (PMID: 28821804) which is considerably below the lowest dose with maximum effect. In contrast, in our comprehensive study we first identified the optimal BTZ-043 treatment dose in BALB/c mice before we assessed its efficacy under conditions of centrally necrotizing granulomas. Importantly, we now also demonstrate a local antimycobacterial activity of BTZ-043 in necrotic granulomas. In addition, we also investigated the potential development of drug resistance. These aspects clearly distinguish our study from previous preclinical reports.

Taken together, our combination of MALDI imaging, measurement of the lesional efficacy and concentration of BTZ-043, we can now indeed make statements about the local relationship between drug concentration and activity in the granuloma. In this context, it should be emphasised that, to our knowledge, the combination of these techniques has never been used in research into new antimycobacterial drugs. Because the medical scientists amongst the authors of the present manuscript also conduct clinical studies on BTZ-043 in TB patients (e.g. a phase IIa trial, Heinrich 2023, DOI:10.2139/ssrn.4601314.), the demonstration of lesion penetration and active killing will substantially influence the design of further clinical trials, as both properties are considered important factors for shortening therapy. This data is even considered important for submission to ethics and regulatory authorities.

Manuscript modification:

These aspects are covered throughout the revised manuscript.

There are also concerns regarding the interpretation of the analytical data.

Response

We have addressed these concerns in our detailed reply below.

General Comments

p. 13 line 237-238: BTZ-043 treatment reduced bacterial burden by 86% does not match the data presented for the log₁₀ CFU reduction of 0.9 presented following 10 days of therapy. How was this % reduction calculated and from where?

Response

We are happy to provide more detailed information on the CFU reduction to answer the question. Before the start of treatment, we determined a mean pulmonary bacterial burden

of 35.1×10^7 CFUs and after 10 days of therapy a burden of 4.8×10^7 CFUs which corresponds to 13.8% of the initial CFU counts. Therefore, the statement in the manuscript that BTZ-043 treatment reduced the bacterial burden by 86% is correct. The determined CFU counts before and after treatment have been included in results part of the revised version of the manuscript. Furthermore, the calculation of the decadic logarithm results in $8.55 \log_{10}$ CFU before and $7.69 \log_{10}$ CFU at the end of BTZ-043 therapy which has been included in the caption of Figure 3.

Manuscript modification:

Line 254: “Since in IL-13^{tg} mice, most pulmonary mycobacteria reside in highly stratified centrally necrotizing granulomas^{10,13} and BTZ-043 treatment reduced the bacterial burden by 86% (from 35.1×10^7 before treatment to 4.8×10^7 after treatment), it is reasonable to assume that BTZ-043 reaches its mycobacterial target within TB lesions.”

Line 1198: “The pulmonary bacterial burden was determined before and after therapy ($8.55 \log_{10}$ CFU and $7.69 \log_{10}$ CFU, respectively).”

BTZ-043 bactericidal activity has previously been demonstrated in cellular and human-like necrotizing lesions during a study using NOS2-deficient mice. This study warrants discussion in the context of the results presented herein.

Response

We agree with the reviewer that Gengenbacher *et al.* published in 2017 (PMID: 28821804) an important study that compared the efficacy of different anti-TB drugs in NOS2-deficient mice which display hypoxic necrotizing lung lesions. However, the authors mainly assessed the bacterial burden after treatment and did not examine drug concentration or distribution. Therefore, we had discussed this study in the context of BTZ-043-mediated CFU reduction in IL-13^{tg} mice in the originally submitted manuscript. In order to emphasize the efficacy of BTZ-043 in all animal models that reflect human TB pathology we thoroughly modified this section of the discussion and furthermore also included the recent study by Ramey (PMID: 37791784) (**lines 513-519**). In addition, by incorporating the recently reported activity of BTZ-043 in TB patients (Heinrich 2023; DOI:10.2139/ssrn.4601314.) we also put emphasis on the translatability of our results to humans (**lines 519-521**). In order to emphasize and compare the activity of BTZ-043 in different mouse models we omitted the comparison to other DprE1 inhibitors in this context (**lines 501-511**).

Manuscript modification:

Lines 501-521: Discussion of our efficacy results in the context of other animal models that also develop human like TB granulomas and likewise demonstrate BTZ-043 activity under conditions of centrally necrotizing granulomas.

p. 15 line 269-272: Evaluation of drug variability would be more suitably carried out using known spiked drug concentrations in tissue mimetics as this would reduce the biological variability observed and provide absolute quantification data on the impact of gamma IR on drug stability. Degradation is structure-dependent for IR and can vary from 0-50+%.

Response

We fully agree that evaluation of gamma-irradiation effects is an important aspect. Therefore, we already targeted this aspect by the analysis of tissue section in our original manuscript. As suggested by the reviewer, we have complemented this approach with an evaluation of spiked tissue mimetics. For this purpose, tissue homogenates were spiked with BTZ-043 concentration similar to those we found in mouse lung sections. These were analyzed by LC-MS/MS and MALDI imaging in order to assess changes in BTZ-043 abundance. As a result, we did not observe a significant impact of gamma-irradiation on the stability of BTZ-043 in these mimetic tissues, based on LC-MS/MS and MALDI imaging. Consequently, we now confirm our previous results which we had obtained by analyzing lung tissue sections that our gamma-irradiation protocol for sterilization of mycobacteria does not lead to significant degradation of BTZ-043.

Manuscript Modifications:

Lines 325-330: discussion of our data obtained in tissue mimetic in the results section.

New Supplementary Figure 3a-b: LC-MS/MS and MALDI imaging measurements of tissue mimetics (parameters for irradiation are described in the caption of Supplementary Figure 3).

Supplementary Methods: the method for “Assessment of impact of gamma-irradiation on BTZ-043 stability” by LC-MS/MS and MALDI imaging is described in detail.

p.16 line 298-300: The left granuloma is described as containing a thicker macrophage layer when this appears to be a mostly cellular or an early necrotic lesion, with what minimal necrosis. Whereas the right lesion has a larger necrotic core and appears to be a more mature necrotic lesion. A significant difference in drug accumulation and mean intensity is observed between these lesions due to the difference in necrosis. These pathologies are important for drug penetrance and efficacy as penetration into cellular lesions is more efficient and Mtb is readily sterilized in these lesions. This sentence could use additional description of granuloma pathology and type in relation to drug penetrance.

Response

A closer inspection of the two granulomas in Figure 5 (formerly Figure 4) revealed that the different structure is mainly due to different regions which have been sectioned. This also explains differences in drug distribution. We have modified the text in order to explain this situation.

Manuscript modification:

Lines 355-360: “The right granuloma is sectioned in its central region showing a prominent necrotic core. The left granuloma is sectioned in a more peripheral region so that the cellular layer which mainly consists of foamy macrophages is more prominent. A neutrophil-rich zone between the macrophage layer and the necrotic core is present in both granulomas.”

p. 16 Fig. 4e-f and description: There is a lot of focus on sublesional penetration but little attempt has been made to analytically characterize the accumulation of BTZ-043 in the cellular vs necrotic regions. Mean intensity is given for the entire inside of the caseum for two lesions with very different pathologies and the data reflects this. It would be beneficial to mark where the cellular macrophage regions end and the necrotic regions begin for the data presented.

Response

To better distinguish the cellular and necrotic sublesional compartments, we have marked these areas in the penetration plots of Figure 5 f (formerly Figure 4f; grey shaded areas) as requested by the reviewer. In order to accommodate this additional information, we have split up Figure 5f in two different diagrams, one for each granuloma (the data points remain unchanged).

Concerning the “mean intensities”, we do not give the mean intensity for the entire inside of the necrotic center. Instead, one data point represents the mean intensity of BTZ-043 for a certain distance from the granuloma edge, i.e. several data points are given for the inside of the necrotic region– for each granuloma separately. We hope that this becomes clearer now by the more detailed presentation in the modified Figure 5f. For more details on this approach you can refer to our recent publication (Kokesch-Himmelreich 2022), which is also mentioned in the text (“penetration analysis tool¹²”).

Manuscript modifications:

Figure 5f: indication of cellular macrophage region and necrotic region and split up data in two separate diagrams.

p. 17 from line 317, Fig 5 and Supplemental Table 3: The data presented in the manuscript and in supplemental table 3 do not correlate. For example, the mean concentrations presented in the supplementary for “a biological replicate” are: 0.5 h = 1.12 ± 0.19 ; 2 h = 1.10 ± 0.25 ; 4h = 1.94 ± 0.37 ; 8h = 0.65 ± 0.06 ng/mg cryosections. The values given in the text and figure are as follows: 0.5 h = 1.32 ± 0.33 ; 2 h = 0.91 ± 0.15 ; 4h = 0.79 ± 0.12 ; 8h = 0.27 ± 0.05 ng/mg cryosections.

Response

We agree with the reviewer that the BTZ-043 concentrations depicted in Figure 6a (formerly Figure 5) of main paper do not correlate with concentrations provided in the Supplementary

Table 3 since these were generated from different animals for each time point and represent therefore inter-animal variations. In more detail, pulmonary BTZ-043 concentrations were determined in a total of 8 Mtb-infected IL-13^{tg} mice. Two mice were analysed at each time point after drug administration (0.5 h, 2 h, 4 h, 8 h) whereby the concentrations obtained from one mouse are depicted in Figure 6a of the main paper and the results obtained for the additional mouse per time point which we term “biological replicate” are shown in the Supplementary Table 3. These details have been added to the results part of the paper (**lines 373-382**), the caption of Figure 6a (**lines 1242-1244**) and the legend of the Supplementary Table 3 of the revised manuscript and supplement, respectively. However, we want to emphasise that our main statement within this context is that the pulmonary BTZ-043 concentration is manifold above the *in vitro* MIC for a prolonged period after drug administration which was demonstrated by providing lung concentrations from two different mice per time point.

Manuscript modifications:

Lines 373-382: Results were rephrased for clarification.

Lines 1242-1244: The caption of Figure 6a contains a more detailed description of samples analysed.

Supplementary Table 3: The legend of the table contains a more detailed description of samples analysed.

The description and methodology do not indicate how many mice were used to generate the cryosection LC-MS data. It appears to be one sample for the figure and one for the supplemental with five technical replicates (5 cryosections from the same sample). Information on biological and technical replicates should be included to ensure experimental and analytical rigor.

Response

We thank the reviewer for bringing our attention to this missing information. The reviewer’s remarks regarding the number of biological and technical replicates are correct. For each time point after BTZ-043 administration we have prepared serial cryosections from a total of 2 mice (biological replicates) and 5 lung cryosections of each mouse (technical replicates) were used for BTZ-043 quantification by LC-MS/MS. This information has been provided in the caption of Figure 6a (formerly Figure 5) and the legend of the Supplementary Table 3 of the revised versions of the manuscript and supplementary information, respectively.

Manuscript modifications:

Lines 1242-1244: “Change of pulmonary BTZ-043 concentration over time. Lung cryosections (n=5) from 1 mouse per time point were prepared for BTZ0-43 quantification by LC-MS/MS which is depicted as [ng/mg] on the left y axis.”

Supplementary Table 3: “Biological replicates of the pulmonary BTZ-043 concentrations shown in Figure 6a. Lung cryosections (n=5)

obtained from an additional mouse at each time point after drug administration were analyzed. “

There is also no data or information on BTZ-043 plasma levels from these time-points or if normalization to plasma concentration was carried out prior to calculating lung concentrations.

Response

We have thought a lot about this comment from the reviewer. However, it has been shown for several antibiotics that the concentration in lesions can be remarkably different from those measured in the plasma and the normalization of the lung concentration to the plasma level is rather unusual in this particular field (PMIDs 27227164, 37791784). Moreover, the plasma concentration is not considered meaningful for the effect of a drug against Mtb in the lesions (PMID 24487820).

Manuscript modification:

Lines 369-370: We would like to refrain from normalizing the tissue concentration of BTZ-043 against its plasma level. However, we explain this special circumstance for TB and give a reference (PMID 24487820).

The concentrations of BTZ-043 are reported to be substantially higher than the MIC at all time-points investigated. This is not supported by the data shown. This data is for cryosections of lung tissue and therefore reflects normal, cellular and necrotic regions and does not provide information on the concentration of BTZ-043 within the necrotic core to correlate with MIC.

Response

The LC-MS/MS data in Figure 6a (formerly Figure 5) is indeed data of cryosections of lung tissue and shows that the “pulmonary BTZ-043 concentrations” are substantially higher than the MIC at all time points investigated. Together with the MALDI imaging experiments, we can conclude that the concentration of BTZ-043 within the necrotic core must be above the MIC at 4 and 8h after the last administration, because a) BTZ-043 concentration in the whole section are at least 100x above the MIC and b) MALDI imaging for these time points shows an accumulation of BTZ-043 in the necrotic area compared to the surrounding tissue.

We now support this assumption by additional experiments using LCM coupled with LC-MS/MS on lung sections from 4h after the last administration. These experiments revealed a mean BTZ-043 concentration of 570 ng/mL within the necrotic core and is therefore manifold above the reported MIC of BTZ-043 (1 ng/mL).

Manuscript modification:

Lines 395-400: We present the results of our LC-MS/MS analysis of laser-capture microdissected lung tissue (Figure 6b).

Lines 530-535: Discussion of these results.

Lines 882-901: We describe the method of LCM of selected lung tissue areas and subsequent LC-MS/MS

p. 19 Fig 6 and description: It is unclear if these images are processed on the same intensity scale across the time course study. This information or the addition of the absolute intensity values for each ion image would enable a more accurate analysis of the accumulation of BTZ-043 at each time post-dose.

Response

Ion images in original Figure 6 (Figure 7 in the revised manuscript) were processed on individual intensity scales. As per the reviewer's suggestion we have included the absolute intensity values of BTZ-043 in the ion images of each post dose time point.

Manuscript modification:

Addition of the absolute values of BTZ-043 in the ion images of each post dose time point in Figure 7 and Supplementary Figure 9. The captions have been amended accordingly.

Specific comments

p. 6 line 80-81: The description of the requirement of a spatial resolution in the range of 10 μm for reliable identification of drug compounds in sub-organ structures is misleading. The average macrophage diameter is $\sim 21 \mu\text{m}$ in diameter, foamy macrophages are larger. Additionally, drug accumulation is rarely cell-specific and more regional-specific. A resolution of 50 μm is sufficient to detect drug accumulation in the center of necrotic regions as well as differential accumulation in the cellular regions of granulomas. As has been shown repeatedly by the Dartois lab (for example, PMID: 30427309).

Response

We agree that not all MALDI imaging studies require high spatial resolution. We would like to argue, however, that the spatial resolution required to reliably identify a differential accumulation in sub-lesion compartments is dependent on the size of the granuloma. The mentioned publication (PMID: 30427309) investigates the distribution of fluoroquinolones in granulomas of rabbit lungs. A reliable identification of drug retention via MALDI imaging in these granulomas can, because of their substantially larger size, be achieved with a spatial resolution of 50 μm . However, for the smaller size of granulomas in the murine model of our study a higher spatial resolution in the range of 10 μm is required. In our original version, we specifically referred to "sub-organ structures in murine tissue". It is also worth noting that, in contrast to rabbits, mouse models allow for more comprehensive efficacy studies to complement the MALDI distribution data.

Manuscript modification:

Lines 109-114: We have rephrased the respective section in order to clarify our point.

p. 15 line 268-269: γ -irradiation is not the only sterilization technique for infected tissue. Whole tissue heat sterilization has been shown for a number of pathogen causing microbes that require biocontainment (PMID: 25966989). More recently, an on-slide heat-sterilization technique was reported for MSI studies of drugs, lipids and metabolites from Mtb infected lung tissue (PMID: 34672552).

Response

It was not our intention to imply that gamma-irradiation is the only available technique for inactivation of mycobacteria. We agree that our wording was misleading. We have clarified our statement in the manuscript and also included the two mentioned publications. We have also explained why we have used our gamma-irradiation protocol: a) it is well established in our laboratory and b) it enables us to inactivate mycobacteria while keeping the sections on dry ice during this procedure, and thus preventing leaking of drugs.

Manuscript modification:

Lines 320-324: Results were rephrased, and the 2 mentioned studies were cited.

Dear reviewers,

thank you very much for your appreciative evaluation of our revised manuscript. Following your further recommendations, we modified the manuscript and our detailed response to your comments is given below (**line numbers refer to the pdf document with tracked changes**).

Based on the requirements and suggestions made by the editor we have made some further modifications and provide additional information:

- In order to present the data on pulmonary BTZ-043 concentrations in a similar way in the main article (Figure 6a) and supplementary information, data previously described in Supplementary Table 3 is now shown as a diagram in Supplementary Figure 8.
- The LC-MS/MS data calculation for Figure 6 and Supplementary Figure 8 (previously Supplementary Table 3) are now provided in detail in excel documents.
 - The excel file “Figure 6a & Sup Figure 8 Data” contains data associated with Figure 6a and Supplementary Figure 8.
 - The excel file “Figure 6b Data” contains data associated with Figure 6b.
- In order to improve the description of our *in vivo* work, the number of mice used for *in vivo* experiments has been clarified further throughout the manuscript.

Reviewer #1:

The revised manuscript of Römpp and colleagues addressed the initial concerns and recommendations. The work is original and important in the field of drug development for tuberculosis and this version is considerably improved.

One minor comment on the line 227-8 of the merged manuscript, the authors claimed that “By 2 weeks of treatment, this expression was already diminished in the BTZ-043 treated animals compared to the vehicle group (Fig. 4b, right panel).” Looking at that panel it doesn’t seem that it is diminished but rather reduced. I would choose a different verb here.

Response

We agree with the reviewer and have modified the sentence accordingly.

Manuscript modification:

Lines 237-239: “By 2 weeks of treatment, this expression was already **diminished** **decreased** in the BTZ-043 treated animals compared to the vehicle group (Fig. 4b, right panel).”

Reviewer #3:

The revision by Römpp and colleagues of the manuscript entitled “The clinical-stage drug BTZ-043 accumulates in murine tuberculosis lesions and efficiently acts against Mycobacterium tuberculosis” has significantly improved the manuscript. The authors carried out extensive additional studies and have further clarified methodological parameters such as sample numbers, CFU calculations and statistics. I commend the authors for their hard work in advancing the impact of their manuscript. The addition of actual drug quant, lesional drug penetration calculations for granulomas with slightly differing pathologies, and an initial evaluation of drug resistance, has all contributed to the improvement of their manuscript. This manuscript is important to help inform on the further development of optimal dosing of BTZ-043 during clinical trial development.

I do have a few comments:

Figure 6 and corresponding discussion: Showing data for **n=1 biological replicate is not analytically sound**. Technical replicates of adjacent cryosections will of course have similar drug concentrations. Statistical tests are usually carried out on the mean of the technical replicates for n=5-10 biological replicates. This is important for population variability.

Response

We thank the reviewer for this comment. Indeed, the indication of significance levels in Figure 6a was misleading as they originate from a previous version, and we apologize for this omission. We have removed the statistical analysis from Figure 6a and have modified the legend of Figure 6 as well as the corresponding methods section accordingly. We would like to emphasize that we do not make a quantitative statement (anymore) on the time-dependence of BTZ-043 concentrations.

Instead, our main statement in the context of drug concentrations in the lung is that “the pulmonary BTZ-043 concentrations reported in our study are substantially higher than the MIC at all time points investigated” (lines 335-337). This statement is based on data shown in Figure 6a and Supplementary Figure 8 (previously Supplementary Table 3) which were obtained from 8 individual mice (n=8 biological replicates) This has been clarified in the revised manuscript. The underlying data for Figure 6a and Supplementary Figure 8 are provided in the excel file “Figure 6a & Sup Figure 8 Data”.

Manuscript modification:

Lines 335-337: “...pulmonary BTZ-043 concentrations reported in our study are substantially higher than the MIC at all time points investigated (in total n=8 mice for data shown in Fig. 6a and Supplementary Fig. 8).”

Lines 827-829: ~~“BTZ-043 concentrations were tested for normality and analyzed by one-way ANOVA followed by Tukey`s post hoc test for multiple comparisons.”~~

Lines 1120-1121: “Data are shown as mean and SD. ~~and statistical analysis was performed by one way ANOVA with Tukey’s posttest (*p=0.0373, **p=0.0080, ns not significant)”~~”

Regarding Figure 6b we state that the data “indicates an accumulation of BTZ-043 in the granuloma compared to the non-necrotic tissue” (lines 341-342) and that “the concentration of BTZ-043 in these necrotic lesions was shown to be manifold above the MIC” (lines 510-511). Necrotic granulomas and non-necrotic tissue were microdissected from serial cryosections that were indeed obtained from 1 mouse (n=1 biological replicate). The information previously given in the figure legend (n=3) refers to the number of samples of each tissue type analyzed by LC-MS/MS measurements (in detail shown in excel file “Figure 6b Data”). This information has been clarified in the revised legend of Figure 6b. These analyses were conducted in response to your comment in the first decision letter which we highly appreciate. As already recognized by the editor, it would be difficult to add more biological replicates to this experiment since suitable cryosections from additional mice with clear pathology (which is required for laser capture microdissection) are not readily available.

However, it is important to note that the statements for Figure 6b are supported by our combination of LC-MS/MS data and MALDI imaging data as both analyses were conducted on cryosections coming from the same lung biopsies. The accumulation of BTZ-043 in necrotic granulomas compared to the surrounding tissue observed 4 h and 8 h after drug administration (MALDI imaging; Figure 7 g, h and Supplementary Figure 10 g, h) in combination with BTZ-043 concentrations determined in nearby lung cryosections (LC-MS/MS, Figure 6a and Supplementary Figure 8) strongly indicates an accumulation of BTZ-043 in necrotic lesions with concentrations manifold above the MIC. These observations were made for a total of 4 different mice (n =4 biological replicates; 2 mice each at 4 h and 8 h after drug administration).

Manuscript modification:

Lines 1127-1128: “Lung cryosections obtained from 1 mouse were used to collect 3 samples of each tissue type for subsequent LC-MS/MS measurements (n=3; color code: ...”

Lines 362-364: “In addition, all granulomas obtained 4 h and 8 h after administration (in total n=4 mice) showed a BTZ-043 accumulation in the necrotic lesions compared to the surrounding tissue.”

The argument the authors present for the requirement of 10 µm or high-resolution imaging based on the size of the granuloma is quite weak. A 30-50 µm pixel resolution of even the smallest granulomas would provide enough information to determine if the drug was able to reach the core of the necrotic center. This is also shown by the authors in their 30 µm pixel

images for the spatial and temporal distribution of BTZ-043 in Figure 7. At or near single cell imaging is only really required in situations where differentiation of neighboring single cells with different populations or phenotypes are required. Or when there truly is a heterogenous accumulation of drug in small cellular regions, etc..

Response

We agree that 10 μm is not “required” for all MS imaging of TB granulomas in mice. It is however useful for more detailed analysis of the sublesional compartments. We have modified the wording according to a suggestion by the editor.

Manuscript modification:

Lines 85-87: “For the murine model used in this study a spatial resolution in the range of 10 μm is ~~required~~recommended in order to the resolve sublesional structures of necrotic granulomas.”